# Upper Limb Capacity, Performance, and Leisure Participation in Children with Unilateral Cerebral Palsy

**DOI:** 10.3390/s25237120

**Published:** 2025-11-21

**Authors:** Manel Abid, Isabelle Poitras, Léandre Gagné-Pelletier, Carole Rigourd, Charles Sèbiyo Batcho, Catherine Mercier

**Affiliations:** 1Center for Interdisciplinary Research in Rehabilitation and Social Integration (Cirris), CIUSSS de la Capitale-Nationale, 525 Blvd Hamel, Quebec City, QC G1M 2S8, Canada; manel.abid.1@ulaval.ca (M.A.); isabelle.poitras.2@ulaval.ca (I.P.); leandre.gagne-pelletier.1@ulaval.ca (L.G.-P.); carole.rigourd@cirris.ulaval.ca (C.R.); charles.batcho@fmed.ulaval.ca (C.S.B.); 2School of Rehabilitation Sciences, Faculty of Medicine, Université Laval, 1050 Avenue de la Médecine, Quebec City, QC G1V 0A6, Canada

**Keywords:** neurodevelopmental disorders, accelerometry, upper extremity, motor activity, activities of daily living, child

## Abstract

**Highlights:**

**What are the main findings?**

**What are the implications of the main findings?**

**Abstract:**

Children with unilateral cerebral palsy (uCP) experience pronounced sensorimotor impairments on one side of the body, impacting upper limb (UL) use and leisure participation. This study compares UL performance in children with uCP and typically developing (TD) children and examines, within the uCP group, the association between UL capacity, UL performance, and leisure participation. Twenty-one children with uCP (age: 11.2 ± 1.8 years) and 30 TD children (age: 10.6 ± 2.1 years) were recruited. UL performance was measured using accelerometry and expressed in activity counts per minute, UL capacity with the Jebsen–Taylor Hand Function Test, and leisure participation with the Children’s Assessment of Participation and Enjoyment questionnaire. The uCP group showed markedly lower performance in their non-dominant side compared to the TD group (*p* < 0.001, ηp^2^ = 0.57), while dominant-side performance showed no significant difference between groups. UL capacity, UL performance, and leisure participation were not correlated (r range: −0.06–0.20). Children with uCP remained active despite UL asymmetry. The lack of significant association between UL capacity, performance, and participation may reflect contextual factors. These findings highlight the importance of complementing accelerometry with measures capturing other dimensions of motor function (e.g., fine motor skills) for a more comprehensive assessment of real-world UL performance.

## 1. Introduction

Cerebral palsy (CP) is a neurodevelopmental disorder resulting from non-progressive disruptions in the brain, affecting 1.5 to 4 per 1000 births globally [1]. It is the leading cause of motor impairments in children and is frequently accompanied by sensory and cognitive deficits [2]. Unilateral cerebral palsy (uCP) is the most common CP subtype (about 44% of cases) [3]. Because uCP is characterized by lateralized sensorimotor impairments, it results in marked upper limb (UL) asymmetry and compensatory overuse of the dominant UL (i.e., the less affected limb) [4,5,6]. Over time, this compensatory pattern can lead to prolonged underuse of the non-dominant UL (i.e., the more affected limb), potentially hindering its functional development [7]. This phenomenon is called developmental disregard [7].

According to the International Classification of Functioning, Disability and Health, three key concepts structure our understanding of functioning: 1-capacity, which refers to what an individual can do in a standardized or clinical environment (e.g., ability to grasp and lift an object during an assessment); 2-performance, which reflects what an individual actually does in their daily life (e.g., frequency of object manipulation with the impaired hand in everyday life); and 3-participation, defined as involvement in life situations (e.g., playing board games with friends) [8]. For children with CP, a discrepancy tends to exist between the capacity of the impaired limb and its performance [9,10]. This gap may have important implications for participation and represents a central issue addressed in the present study.

Children with CP often encounter reduced participation in daily and leisure activities, which are essential for their development and social inclusion [11,12,13]. Compared to typically developing (TD) peers, they participate in fewer and less vigorous activities, often choosing passive, home-based pursuits [14,15,16,17]. Participation challenges in children with CP result from the complex combination of personal (e.g., capacity, performance) and environmental factors that can either facilitate or hinder their involvement [17,18]. While UL capacity and performance are crucial for daily and leisure activities such as self-care, play, and structured physical activities, their impact on participation in children with CP is underexplored [18]. Standardized clinical assessments help evaluate motor capacity; however, they may not reflect the child’s spontaneous UL performance in daily activities [10,19]. An alternative way to assess UL performance is through accelerometry [20]. It has been demonstrated that accelerometers can be a feasible and reliable method to quantify both more and less affected ULs use in children with uCP [21,22]. However a recent study showed that scores obtained from clinical tools are not significantly associated with the actual use of the more affected arm during daily activities in children with uCP [23], highlighting the importance of accelerometry in reflecting everyday motor behavior [23,24].

Previous research has examined the relationship between UL capacity and UL performance using accelerometry [22,23], or between UL capacity and participation [25,26,27], but none addressed all three dimensions concurrently. Moreover, the specific relationship between UL performance and leisure participation has not yet been investigated. To the best of our knowledge, only one study [28] has investigated these links in children with CP, using standardized measures of gross motor capacity, motor performance (assessed through general physical activity), and participation in daily life. The findings showed that motor performance had a stronger association with participation, whereas capacity was a weaker predictor [28]. Gaining insight into these relationships may support the design of targeted interventions that promote more inclusive and meaningful participation in daily and leisure activities, while encouraging more balanced and functional use of both ULs.

This study had two main (clinical) objectives:1.To compare UL performance, measured by the intensity of UL activity, between children with uCP and TD children and between sides (dominant and non-dominant). We hypothesized that children with uCP would show reduced overall UL activity for both ULs, as well as a greater interlimb asymmetry compared to TD children.2.To explore the associations between UL performance, UL capacity, and their participation in leisure activities outside the school setting in children with uCP. We hypothesized that higher UL capacity and UL performance would be associated with higher leisure participation.

Two secondary (methodological) objectives were defined in relation to the main objectives:1.To examine whether a 2-day (weekend) measurement period provides a reliable estimate of UL performance compared to 5-day (weekdays) and 7-day (full week) periods, and to evaluate the reproducibility of children’s activity patterns across these different recording durations. This validation was only performed within the TD group, because the data for the uCP participants were limited to the weekend recordings. This limitation was explained by their participation in a separate longitudinal study with multiple evaluations of the effects of bimanual therapy, where the use of accelerometry was limited to weekends to ease the burden on both children and their families before, during, and after the intervention.2.To assess the effect of age on the UL performance, given the large heterogeneity in our group in terms of age.

## 2. Materials and Methods

### 2.1. Study Design and Ethics

This cross-sectional observational study was approved by the institutional review board [RIS board: #2020-1961, #2023-2623, and #2023-2684 Centre intégré universitaire de santé et de services sociaux de la Capitale-Nationale (CIUSSS-CN)] in Québec, and the legal guardian of each child provided written informed consent prior to participation. Data for each group (children with TD and with uCP) were acquired prospectively in distinct studies, which accounts for some differences in the data collection described below.

### 2.2. Participants

Recruitment was conducted through the Université Laval mailing list for both groups and by consulting health records of the CIUSSS-CN for the group with uCP. For the uCP participants, the inclusion criteria were as follows: (1) being aged between 7 and 16 years; (2) having a diagnosis of uCP; and (3) having mild to moderate impairments based on the Manual Ability Classification Scale [MACS], i.e., a score between level I and III. The exclusion criteria were defined as follows: (1) having a significant visual or cognitive impairment that could hinder the understanding of the questionnaire assessing leisure activities participation, or the ability to understand the instructions during the motor capacity assessment. TD participants were included if they met the following criteria: (1) were aged 7 to 16 years; and (2) had no developmental conditions or other diagnoses affecting the arms.

### 2.3. Study Protocol

Children with uCP wore wrist accelerometers for 2 days over a weekend, underwent the Jebsen-Taylor Hand Function Test (JTHFT) during a laboratory visit, and completed the Children’s Assessment of Participation and Enjoyment (CAPE). Accelerometry was limited to weekends as part of a broader longitudinal study investigating intensive bimanual therapy across four timepoints (pre, during, post, and 6-month follow-up). To reduce participant burden and ensure adherence across repeated measures, wearing the devices for 7 consecutive days at each timepoint was considered impractical. Children from the TD group underwent a single full week assessment with wrist-worn accelerometers. This longer period of monitoring allowed us to examine whether the weekend recordings provided results comparable to those obtained over 5 weekdays and over the full week period. Moreover, previous research has shown that short monitoring periods of two consecutive days can yield reliable estimates of daily physical activity and gait performance in children with CP, with intraclass correlation coefficients (ICC) ranging from 0.70 to 0.98 when using wearable inertial sensors [29].

### 2.4. Instruments

#### 2.4.1. Accelerometry Data Collection and Processing

Both verbal and written instructions were provided to parents regarding the proper procedure for wearing the wrist accelerometers. Participants were instructed to wear the accelerometers from the moment they woke up in the morning until bedtime, but to remove the devices during water-based activities, such as showering or swimming.

The UL performance was initially measured using the ActiGraph GT9X Link (ActiGraph Corporation, Pensacola, FL, USA; 35 × 35 × 10 mm, 14 g). However, due to the limited number of available units and the discontinuation of this model during our data collection, we were unable to continue using only this device. Since the manufacturer did not guarantee continued availability and support, we opted to transition to the Axivity AX3 (Axivity Ltd., Newcastle upon Tyne, UK; 23 × 32.5 × 7.6 mm, 11 g), which allowed for raw data collection comparable to the ActiGraph. To guarantee comparability between the two accelerometer models used in this study (ActiGraph GT9X Link and Axivity AX3), a pilot validation was carried out in our laboratory. Seven participants wore both devices simultaneously on the same wrist over a 2-day recording period during daily activities. To ensure methodological consistency, all data were analyzed using the same custom MATLAB scripts that process raw acceleration signals in the same way, regardless of the device. Excellent agreement between ActiGraph and Axivity outputs was indicated by the ICC (single measures, consistency) of 0.981 (95% CI [0.893–0.997], *p* < 0.001). With a mean bias of 181.4 AC/min (SD = 186.3) and 95% limits of agreement ranging from −183.7 to 546.6 AC/min, Bland–Altman analysis further validated the good agreement. These results confirmed that both devices offer comparable measurements of UL activity under the same circumstances, which is in line with an earlier comparison between these two devices by Buchan et al. (2022) [30].

Raw accelerometer data were collected using two types of sensors: the ActiGraph GT9X and the Axivity AX3. ActiGraph data were downloaded using ActiLife software (version 6.13.4, ActiGraph Corp., Pensacola, FL, USA), and Axivity data were downloaded using the Open Movement software OMGUI (version 1.0.0.45, Open Lab, Newcastle University, UK). Data were processed offline using a custom MATLAB script (version 9.11, R2021b, The MathWorks Inc., Natick, MA, USA), based on the open-source algorithm developed by Poitras et al. (2020) [31]. Following the approach described by Poitras et al. (2020), a continuous 8th-order bandpass filter was optimized using the developed algorithm in MATLAB to replicate the frequency response of the commercial activity count filter [31]. The final continuous parameters were discretized at 100 Hz to match the sampling rate used in this study. This method ensures numerical stability and allows for accurate filtering of high-frequency motion data. The script applied an eighth-order bandpass filter to remove noise and converted raw signals into 1 s epochs.

The following formula was used to calculate the vector magnitude for each epoch:$$AC=\sqrt{\left\{s_{\left\{xi\right\}}^{2}+s_{\left\{yi\right\}}^{2}+s_{\left\{zi\right\}}^{2}\right\}}$$

The filtered acceleration signals on the *x*, *y*, and *z* axes are represented by $s_{xi},\;s_{yi}$, and
$s_{zi}$, respectively. The total activity counts (AC) were then summed over the entire wear period for each UL, and the mean AC per minute was obtained by dividing the total AC by the total number of minutes of valid wear time.

Visual inspection and manual handling of the data were performed to exclude non-wear periods and sleep. This consistent approach allowed for comparability between the two types of sensors, as both were analyzed using the same processing pipeline applied to raw data.

A recording day was defined as valid when the accelerometer was worn for a minimum average duration of 6 h per day across the two weekend days. This criterion is consistent with previous research on children with CP using ActiGraph [32].

#### 2.4.2. The Jebsen-Taylor Hand Function Test (JTHFT)

The JTHFT was used to evaluate UL capacity in children with uCP [33]. It tests various activities, including card turning, grasping small common objects, simulated feeding, playing checkers, and lifting large, light, and heavy objects [33]. Each activity is timed, and if a task is not completed within 120 s, the test is stopped, and the maximum time of 120 s is recorded for that item. The total time to complete all tasks is calculated separately for each UL. Assessments were consistently administered, beginning with the non-dominant side and then the dominant side. JTHFT was administered by three experienced evaluators (two occupational therapists and one physiotherapist). All evaluators received training and followed the same standardized administration procedures to ensure methodological consistency and minimize inter-rater variability across sessions. JTHFT has been shown to be a valid and reliable tool for assessing manual dexterity in children with CP aged 6 to 18 years [34]. It has excellent internal consistency, with Cronbach’s alpha values of 0.94 and 0.91 for the non-dominant and dominant sides, respectively [34].

#### 2.4.3. Children’s Assessment of Participation and Enjoyment (CAPE)

The CAPE questionnaire consists of 55 items designed to assess the participation of children and adolescents (aged 6 to 21 years) in leisure activities outside the classroom setting [35]. The CAPE provides information on five dimensions of participation in leisure and recreational activities: activity context (where and with whom), diversity, intensity, and perceived enjoyment. The dimension of interest in this study was participation intensity, measured on a 7-point scale ranging from 1 (once in the past 4 months) to 7 (once a day or more). CAPE intensity scores were calculated for five activity types: recreational activities (12 activities), active physical activities (13), social activities (10), skill-based activities (10), and self-improvement activities (10). An overall intensity score can also be computed. The CAPE is widely used in research involving children with CP [36] and shows acceptable psychometric properties among TD children, with internal consistency from 0.32 to 0.76 and test–retest reliability (ICC = 0.67–0.86) for diversity and intensity scores [37].

### 2.5. Statistical Analyses

Statistical analyses were conducted using SPSS Statistics version 30.0 (IBM, Armonk, NY, USA). Descriptive statistics were generated to summarize demographic and clinical characteristics, Chi-square or Fisher’s exact tests were used to assess group differences for categorical variables, and Student’s *t*-tests or Kruskal–Wallis tests were applied for continuous variables depending on data distribution.

To address the primary objective #1, a two-way analysis of covariance (ANCOVA) was conducted to compare the UL performance (in AC per minute) between sides and groups, with age as a covariate. Assumptions of normality (TD group: Shapiro–Wilk W = 0.941, *p* = 0.096 for the dominant side; W = 0.958, *p* = 0.271 for the non-dominant side; group with uCP: Shapiro–Wilk W = 0.968, *p* = 0.697 for the dominant side; W = 0.949, *p* = 0.333 for the non-dominant side), homogeneity of variance (Levene’s test: F = 0.029, *p* = 0.866 for the dominant side and F = 0.962, *p* = 0.332 for the non-dominant side), homogeneity of regression slopes (Group × Age interaction: F = 0.4, *p* = 0.53 for the dominant side; F = 1.313, *p* = 0.258 for the non-dominant side), and linearity between the dependent variable and the covariate were verified before conducting the ANCOVA. The inter-subject factor was the group (Ucp Vs. TD children), and the intra-subject factor was the side (Dominant Vs. Non-Dominant). Post Hoc comparisons were performed between groups for side and between sides within each group. Bonferroni corrections were applied to the *p*-values to adjust for multiple testing, and the corrected *p*-values are reported. Effect sizes were calculated using partial eta squared (η_p_^2^) and interpreted as small (0.01–0.059), medium (0.06–0.13), and large (≥0.14) [38]. To address the primary objective #2, correlation analyses were performed to test the relationship between UL performance (AC per minute for each side), UL capacity (overall JTHFT for each side), and participation intensity in leisure activities (overall score of CAPE intensity) in children with uCP only. Spearman’s rank correlation analyses were performed, as JTHFT scores for both dominant and non-dominant sides did not meet the assumption of normality (Shapiro–Wilk *p* < 0.05). For the JTHFT, raw completion times (in seconds) were used for each hand without age normalization. For the CAPE, only the overall intensity score was included in the correlation analyses, representing the intensity of participation across all activity types. Correlation coefficients were interpreted using standard benchmarks: small (r ≥ 0.3), moderate (r ≥ 0.5), and large (r ≥ 0.6) [39]. To address the first secondary objective focusing on methodological aspects #1, paired *t*-tests and ICC (single measures, consistency) were performed to compare weekend, 5 weekdays, and full week measurement periods for each side in the TD group. ICC values below 0.50 indicate poor reliability, values between 0.50 and 0.75 indicate moderate reliability, values between 0.75 and 0.90 indicate good reliability, and values above 0.90 indicate excellent reliability [40]. Scatterplots with the line of identity and Bland–Altman plots were also generated to assess inter-individual variability and agreement across periods. To address the second secondary objective focusing on methodological aspects #2, Pearson correlation coefficients were computed to evaluate the relationships between age and UL performance for each side within each group (uCP and TD). All between-group comparisons were based on the 2-day weekend monitoring data. The longer monitoring periods (5 weekdays and 7 days) available for the TD group were used solely for secondary analyses assessing the reproducibility of weekend measurements relative to longer durations. Statistical significance was set at α < 0.05 for all analyses.

## 3. Results

### 3.1. Sample Description

Four TD participants were excluded because their average wear time was below 6 h per day across the two weekend days. The final sample, for which there was no missing data, included 30 subjects in the TD group and 21 subjects in the uCP group. For the TD group, most of the participants were males (62.1%; not significantly different from the uCP group, *p* = 0.59), with a mean age of 10.6 ± 2.11 years (not significantly different from the uCP group, *p* = 0.41), and 86.7% were right-handed. Dominance differed between groups (*p* = 0.007, which was expected given the balanced number of uCP participants for whom the more affected side was the right side vs. the left side). The mean wear time for the TD group was 10.2 ± 2.76 h per day (not significantly different from the uCP group, *p* = 0.284). Most TD participants (93.3%) wore the ActiGraph device (not significantly different from the uCP group, *p* = 0.11). Detailed individual and group data for the uCP group are presented in Table 1.

### 3.2. Main Results

#### 3.2.1. Comparison of UL Performance Between Sides and Groups

A post hoc sensitivity power analysis conducted in G*Power (version 3.1; α = 0.05, two-tailed, 1 − β = 0.80) based on the final sample sizes (*n* = 21 for the uCP group and *n* = 30 for the TD group) indicated that the study was powered to detect between-group effects of Cohen’s d ≈ 0.81 (η_p_^2^ ≈ 0.14), corresponding to large effect sizes. Figure 1 shows the comparison of UL performance (AC per minute) between groups and sides. After adjusting for age, the ANCOVA showed significant main effects of both side (95% CI [−1192; −380] for non-dominant vs. dominant side, F = 5.129, *p* = 0.028, η_p_^2^ = 0.097) and group (95% CI [−1342; −425] for Ucp Vs. TD group, F = 15.032, *p* < 0.001, η_p_^2^ = 0.238). Furthermore, a main effect of age was identified (F = 11.926, *p* = 0.001, η_p_^2^ = 0.199). The interaction between group and side was significant (F = 84.075, *p* < 0.001, η_p_^2^ = 0.637) after adjusting for age. Multiple comparison analyses revealed a significant difference between groups for the non-dominant side (95% CI [−2150, −1278], F = 62.41, *p* < 0.001, η_p_^2^ = 0.565), but not for the dominant side (95% CI [−597, 491], F = 0.038, *p* = 0.846, η_p_^2^ = 0.001). A significant difference between sides was found for the uCP group (95% CI [−2120, −1563], F = 176.737, *p* < 0.001, η_p_^2^ = 0.786), but not for the TD group (95% CI [−413, 52], F = 2.436, *p* = 0.125, η_p_^2^ = 0.048). Effect sizes were large for all significant comparisons, except for the main effect of side, which was moderate.

#### 3.2.2. Correlation Between UL Performance, UL Capacity and the Participation Intensity in Leisure Activity

As summarized in Table 2, no significant associations were found between UL performance and UL capacity, between UL performance and participation intensity, or between UL capacity and participation intensity for either the non-dominant side or the dominant side. A Post Hoc power analysis indicated that, with the sample size of 21 participants in the uCP group, the study had limited statistical power (approximately 0.06–0.28) to detect small-to-moderate correlations between UL performance, capacity, and participation intensity.

### 3.3. Results of the Secondary Objectives Focusing on Methodological Aspects

#### 3.3.1. Reproducibility of Weekend and Longer Monitoring Periods of UL Performance

Analyses were conducted on the 28 TD participants who completed valid 7-day recordings. Paired *t*-tests showed no significant differences in AC per minute between the weekend and full week periods for either side (Table A1). Similarly, no significant differences were observed between the weekend and 5 weekdays periods for the non-dominant side or the dominant side (Table A2). ICC indicated moderate consistency between weekend and full week periods, as well as between weekend and five-weekday periods, for both sides. The scatterplots indicated that the weekend period generally reflects the behavior observed over the full week (Figure A1) or the 5 weekdays (Figure A2). However, the dispersion of points around the line of identity highlights inter-individual variability, indicating that this concordance is not perfect. The Bland–Altman analysis showed small mean biases across comparisons (≈153–221 AC per minute), suggesting no systematic over- or underestimation of measurements during the weekend period relative to longer monitoring periods (Figure A3 for full week vs. weekend and Figure A4 for 5 weekdays vs. weekend). Only one to two children were outside of the 95% limits of agreement for each comparison, showing that differences were minimal for most participants.

#### 3.3.2. Effect of Age on UL Performance for Each Group

This analysis included all participants (*n* = 21 uCP; *n* = 30 TD) with complete, valid data. In the uCP group, AC per minute did not significantly correlate with age for either the non-dominant (r = −0.43, *p* = 0.054) or the dominant (r = −0.3, *p* = 0.18) side. However, in the TD group, AC per minute was significantly and negatively correlated for both the non-dominant (r = −0.5, *p* = 0.004) and the dominant (r = −0.46, *p* = 0.011) sides, suggesting that older children generally showed lower UL performance. The scatterplots are presented in Appendix B (Figure A5).

## 4. Discussion

To our knowledge, this is one of the first studies comparing UL daily performance, assessed through accelerometry, between sides in children with and without uCP and to explore the relationships between UL capacity, UL performance, and participation in leisure activities. The results showed a significant asymmetry in the uCP group, indicating a reduced use of the non-dominant limb. Performance of the dominant limb, however, did not differ significantly between groups. Furthermore, no significant correlations were observed between capacity, performance, and participation in the uCP group.

In children with uCP, the use of the non-dominant hand in daily life represented, on average, 57% of the AC per minute observed on the dominant side. This is consistent with previous studies documenting underuse of the more affected side [21,24,41,42], and could be explained by the phenomenon of developmental disregard. This phenomenon is believed to result from unfavorable learning experiences, such as a lack of positive reinforcement [43], increased cognitive load [44], or incomplete neurological development of motor circuits [45]. Over time, this may lead the dominant UL to assume a compensatory role in most daily tasks. While this interpretation remains speculative, it is consistent with evidence suggesting adaptive motor strategies that sustain overall activity despite unilateral limitations [21,24,41,42]. This could also explain why, in our results, the dominant side did not significantly differ between uCP and TD children. These findings emphasize the need to prioritize rehabilitation strategies that specifically promote the use of the more affected limb. Evidence-based interventions such as constraint-induced movement therapy and bimanual intensive therapy have been shown to enhance motor function and reduce learned nonuse by intensifying practice engaging the more affected arm [46,47]. In addition, incorporating motivational approaches such as playful, goal-directed activities or gamified environments may further encourage spontaneous use of the affected arm in daily life [48,49]. Integrating these approaches could help reduce developmental disregard and promote more balanced use of both arms in children with uCP [48,49].

Moreover, the absence of a significant difference between groups in the dominant UL performance suggests that children with uCP maintain similar levels of overall activity to their TD peers. This is consistent with a recent study [24] that reported no significant difference for dominant limb use between groups, despite asymmetrical use of the non-dominant UL in uCP children. Similar patterns have been reported in studies examining global physical activity behaviors [50,51]. For instance, a recent study using thigh-worn accelerometry found that young children with CP aged 6–12 years do not have different physical behaviors than their TD peers [50]. Similarly, another study using wrist-worn monitors found no significant differences in moderate physical activity or step count between groups in the same age range [51]. In contrast, earlier studies that used hip- or waist-worn accelerometers [52,53,54] consistently reported reduced overall physical activity in children with CP, including more sedentary behavior and fewer bouts of moderate to vigorous physical activity. These findings suggest that physical activity levels may be highly context-dependent, influenced by personal and environmental factors [18], as well as by differences in accelerometer placement. Therefore, generalizations about inactivity in this population should be made cautiously.

While UL capacity and UL performance are often assumed to play a central role in supporting children’s participation in everyday life, our findings revealed no significant correlation with leisure participation. This disconnect may arise from differences in what each measure captures: the JTHFT assesses fine manual dexterity [33], whereas accelerometry evaluates the intensity of proximal movements (i.e., more gross motor function), without considering precision or coordination [20]. Another possible explanation involves cortical reorganization following early brain lesions in uCP [55,56]. Due to the high plastic potential of the developing brain, motor control of the more affected UL can be reorganized either within the contralesional (unaffected) hemisphere or through spared regions of the ipsilesional (affected) hemisphere [55,56]. This adaptive reorganization enables children with uCP to perform movements via alternative neural pathways [56,57]. However, these compensatory mechanisms often favor functional efficiency over precise motor control, which may result in relatively preserved gross motor abilities but persistent impairments in fine motor skills [56,57]. Such neuroplastic changes could therefore mask the relationship between capacities assessed through clinical tests and real-world performance as well as leisure participation. Meanwhile, the CAPE questionnaire measures participation intensity in various leisure activities, from fine-motor tasks like performing crafts to more gross-motor ones like hiking [35]. This diversity of activities in which UL is involved makes it difficult to establish a direct link between UL capacity, UL performance, and participation, especially given that each child engages in a unique set of activities. This could be explained by the fact that leisure involvement is driven more by personal and environmental factors, including personal preference, motivation, enjoyment, self-confidence, family support, and accessibility are more critical than motor capacity and performance [18]. The lack of direct associations among UL capacity, UL performance, and participation may further suggest the existence of more complex, non-linear interrelationships. All factors, including psychosocial and environmental, such as perceived competence and family support, would possibly affect the ability of motor capacity and performance to translate into real-life engagement [58,59]. Future studies should consider analytical approaches such as structural equation modeling or path analysis to better explain the complex determinants of participation in children with uCP [58,59]. These findings contrast with studies in post-stroke adults, where significant correlations between capacity, performance, and participation have been reported [60,61,62,63,64]. This may reflect fundamentally different developmental trajectories: children with uCP have never had symmetrical function and develop compensatory strategies early on, while adults post-stroke experience a loss of previously acquired bilateral motor skills [41,65]. This might lead to a tighter link between what adults can do, what they actually do, and how they engage with their environment. Moreover, children’s participation is often motivated by enjoyment and social context, which override difficulties related to motor impairments [66]. However, adults usually prioritize functional activities to regain their autonomy, making these relationships more evident [67].

### 4.1. Limitations

This study has some methodological limitations. First, data from uCP and TD participants were obtained from independent studies, which explains some differences in the data collection protocols. Comparisons between groups on UL performance were based on monitoring over two weekend days. To address this limitation, we evaluated the validity of weekend measurements in the TD group by comparing them with both 5 weekdays and full week recordings. At the group level, mean UL performance did not significantly differ between periods, but our results indicated only moderate-to-good correlations and revealed some inter-individual variability. According to these findings, even though a weekend period offers a reasonable approximation of longer monitoring durations at the group level, interpreting individual-level results should be done cautiously. Daily activity levels can fluctuate considerably from one day to another or across seasons, and it is therefore possible that the short monitoring period did not fully capture the diversity of activities reported in the CAPE. Furthermore, external factors such as weather conditions, weekend routines, or family schedules may also have influenced activity levels during the weekend. Second, the age range of participants was large, potentially influencing UL performance. Indeed, older TD children showed significantly lower activity. Although no such effect was seen in the uCP group, we statistically controlled for age in the group comparison by including it as a covariate. Third, another limitation of the study is the use of two different accelerometer models for UL performance measurement. However, this was addressed by ensuring that both devices recorded raw acceleration data, and all data were processed using the same analysis pipeline developed in MATLAB prior to the study. The validity of combining data across devices was supported by comparative tests carried out prior to data collection, which showed no significant differences between devices (Wilcoxon *p* = 0.078), excellent reliability (ICC = 0.98, 95% CI [0.89–0.99], *p* < 0.001), and minimal bias in Bland–Altman analysis (mean bias = 181.4 AC/min; limits of agreement: −183.7 to 546.6 AC/min). Another limitation concerns the relatively small sample size in the uCP group (*n* = 21), which reflects the well-known challenges of recruiting this clinical population. This limited number of participants may have reduced the statistical power to detect moderate correlations. A Post Hoc power analysis confirmed that, with this sample size, the study was sufficiently powered to detect only large effects (Cohen’s d ≈ 0.81), suggesting that smaller or moderate effects might not have been captured. The study’s small sample size resulted in limited statistical power for detecting moderate correlations between UL performance, capacity, and participation intensity, as confirmed by the Post Hoc analysis. Non-significant results should therefore be interpreted with caution. The relatively small and potentially unbalanced sample in terms of sex distribution and severity levels may have influenced the generalizability of our findings. Finally, only children with uCP presenting with mild to moderate manual ability limitations (MACS I–III) were included in our sample. As such, the findings might not accurately reflect the larger uCP population.

### 4.2. Recommendations

Accelerometry offers an ecologically valid method to monitor how children use their ULs in their natural environment, rather than during isolated tasks during clinical tests [68]. By continuously recording activity over multiple days, this approach provides an enhanced representation of motor behavior, free from examiner bias [69]. This is particularly relevant for populations living with hemiparesis, for whom accelerometers are particularly useful in detecting asymmetries in limb use and adaptive strategies that may remain unnoticed during short, structured assessments [68,70]. Although accelerometry offers important insights into the amount and symmetry of UL use, it does not assess qualitative aspects of movement [71,72]. This drawback points out the necessity for additional technologies capable of capturing fine motor skills in real-world contexts. Advancements in sensor fusion and multimodal motor assessment further expand the potential to overcome accelerometry’s inherent limitations [68]. Merging accelerometers with other sensing technologies, such as gyroscopes, markerless systems, and pressure or force sensors, allows assessment of both the kinematic and kinetic features of UL movement [68]. Sensor-integrated gloves show a potential innovation that provides a thorough evaluation of hand use by accurately monitoring finger movements and object manipulation [73]. The combination of these complementary technologies into unified sensor frameworks has a greater potential for the ecological validity and clinical sensitivity of UL assessments in children with uCP. From a therapeutic perspective, integrating real-world activity measures into rehabilitation programs may advance individualized care by identifying distinct patterns of limb use [46,47]. Tools such as accelerometry and multimodal sensors allow clinicians to identify underused limbs and modify treatment strategies, including bimanual intensive therapy or devices providing feedback regarding limb use, to promote more balanced and efficient UL involvement in everyday situations [68,74].

## 5. Conclusions

This study highlights the pronounced asymmetry in daily UL performance within children living with uCP. However, the performance of the dominant side did not differ from that of TD peers, suggesting that compensatory patterns might allow for maintaining overall activity levels regardless of motor impairments. Accelerometry offers valuable features for objectively quantifying daily UL use but does not offer information on qualitative aspects such as fine motor skills. This emphasizes the need to complement accelerometry with additional tools capable of assessing both qualitative and quantitative dimensions of movement. Applying these methodological advancements is essential for clinical use, as combining real-world activity assessments, such as accelerometry and sensor-integrated gloves, with standard clinical measures can more effectively guide personalized therapy and detect significant changes in daily UL function in children with uCP. Our findings suggest that while accelerometry provides objective insights into daily UL activity, participation outcomes in children with uCP might depend more on contextual and psychosocial than on motor factors.

## Figures and Tables

**Figure 1 sensors-25-07120-f001:**
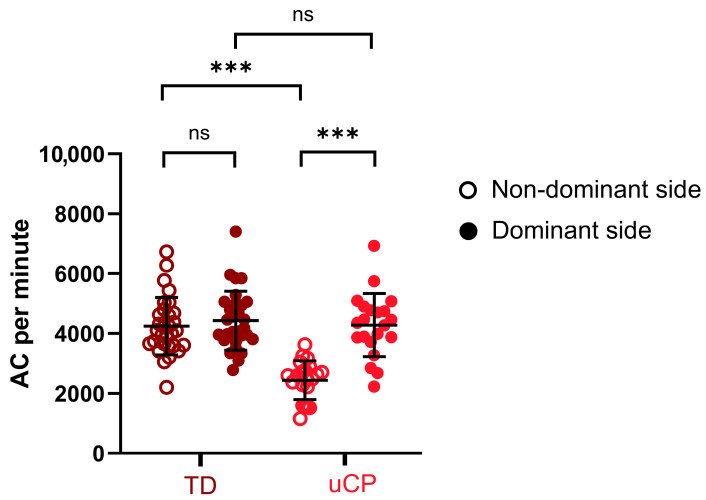
Mean and comparison of UL performance (activity counts (AC) per minute) between the non-dominant and dominant sides across the two groups. AC per minute: activity counts per minute; uCP: children with unilateral cerebral palsy; TD: typically developed children; UL: upper limb. Brown circles represent the activity count data of the TD group. Red circles represent the activity count data of the uCP group. ns: non significant; ***: *p* < 0.001; Error bars: mean and SD.

**Table 1 sensors-25-07120-t001:** Individual data of the CP group.

Subject	Age (Years)	Sex	Affected Side	MACS Level	Device Used	Mean Wear Time (Hours/Day)	AC per Minute		JTHFT Scores (s)		CAPE Score
							ND	D	ND	D	
S1	11	F	R	2	ActiGraph	11.58	1596	4274	215	31	2.8
S2	14	M	R	2	ActiGraph	13.97	1158	2230	46	26	1.6
S3	11	F	L	3	ActiGraph	13.18	3254	6929	549	36	2.4
S4	14	M	L	2	ActiGraph	7.10	2283	4351	83	34	2.2
S5	12	M	R	3	ActiGraph	11.73	2769	5743	510	26	1.7
S6	9	F	L	2	ActiGraph	10.88	2641	4902	385	33	2
S7	9	M	R	1	ActiGraph	12.09	2381	2676	47	44	1.9
S8	13	M	L	1	ActiGraph	12.22	2221	2848	37	24	2.3
S9	11	M	L	2	ActiGraph	10.22	2598	3883	96	32	2.7
S10	7	F	L	3	ActiGraph	9.64	2886	5098	273	42	3.2
S11	9	F	R	3	ActiGraph	10.90	2719	5077	528	43	2.7
S12	13	M	L	1	ActiGraph	10.66	3047	3725	37	32	2.4
S13	10	F	L	1	ActiGraph	12.56	2594	3869	87	45	2.6
S14	11	F	R	3	ActiGraph	10.10	1522	3278	576	33	2.9
S15	8	M	R	3	Axivity	13.61	2479	4744	590	38	3.3
S16	12	F	L	1	Axivity	7.17	2472	4471	92	42	3.4
S17	7	M	R	1	Axivity	12.43	2702	4775	125	41	3.3
S18	15	M	L	3	ActiGraph	11.80	1534	3899	604	166	1.8
S19	13	M	R	2	ActiGraph	10.17	3181	3986	318	30	2.1
S20	14	F	R	3	Axivity	11.15	1582	4702	720	35	1.3
S21	10	F	L	3	Axivity	11.62	3633	4466	540	102	3.7
Mean (SD)/%	11.1 (2.4)	F: 47.6%	L: 52.38%	I: 28.6%; II: 28.6%; III: 42.9%	Axivity: 23.8%	11.18 (1.77)	2441 (647)	4282 (1055)	308 (238)	45 (32)	2.5 (0.7)

Note: F: female; M: male; R: right; L: left; MACS: Manual Ability Classification System; ND: non-dominant arm; D: dominant arm; JTHFT: Jebsen-Taylor Hand Function Test; AC: activity counts; CAPE: Children Assessment of Participation and Enjoyment Questionnaire.

**Table 2 sensors-25-07120-t002:** Summary of correlations between UL performance, UL capacity, and participation intensity in children with uCP.

Correlation Tested	Side	r	*p*
Performance (AC/min) ↔ Capacity (JTHFT)	ND	0.058	0.801
	D	0.198	0.39
Performance (AC/min) ↔ Participation intensity (CAPE)	ND	0.315	0.164
	D	0.211	0.36
Capacity (JTHFT) ↔ Participation intensity (CAPE)	ND	0.051	0.826
	D	0.349	0.121

Note. Correlations computed using Pearson’s r. AC/min = activity counts per minute; JTHFT = Jebsen–Taylor Hand Function Test; CAPE = Children’s Assessment of Participation and Enjoyment; ND = non-dominant; D = dominant.

## Data Availability

The data presented in this study are available on request from the corresponding author. The data are not publicly available due to restrictions related to ethical approval.

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
