# Peer review of "Upper Limb Capacity, Performance, and Leisure Participation in Children with Unilateral Cerebral Palsy"

_sensors, 2025, doi:10.3390/s25237120_

Round 1
Reviewer 1 Report
Comments and Suggestions for Authors
This is a well-executed and timely study exploring how upper-limb performance and capacity relate to leisure participation in children with unilateral cerebral palsy (uCP). The topic is clinically meaningful and fits nicely within Sensors’ interest in applying technology to rehabilitation contexts. The study design is solid, the methods are clearly explained, and the discussion connects well with the broader literature.
That said, a few sections could be tightened to improve clarity and flow, mainly in the introduction and discussion. Adding a little more detail about the accelerometer procedures and emphasising the clinical relevance of the findings would make the paper even stronger.
Abstract
-
Include key numerical findings (e.g., correlation coefficients or effect sizes) to make the results more tangible.
-
In the conclusion, use slightly softer wording, e.g., “no significant association was found” instead of “no association”, to reflect the exploratory nature of the analysis.
Introduction
-
The introduction could be shortened slightly by merging overlapping paragraphs around lines 70–100.
-
It would help to explain why this link between upper-limb use and participation hasn’t been widely examined before (for instance, due to limitations of clinical tools or the variability of leisure activities).
-
Consider adding one or two recent accelerometry studies for context (e.g., Beani et al., 2025; Coker-Bolt et al., 2016).
Methods
-
Provide a short explanation of how you handled calibration between the two accelerometer models.
-
The shorter recording period for the uCP group (2 days vs 7 days) should be justified earlier in the Methods, not only in the Limitations.
-
Clarify whether a single examiner conducted all the JTHFT tests to control for rater variability.
Results
-
Simplify presentation by moving some detailed numeric results to supplementary material.
-
Add a correlation summary table so that key relationships are easier to see at a glance.
-
Ensure all figure legends define abbreviations (e.g., D = dominant, ND = non-dominant)
Discussion
-
Strengthen the mechanistic explanation. Why might no association have been found? Consider discussing compensatory strategies, cortical reorganisation, or task-specific motor control.
-
Expand on the ecological validity of accelerometry compared with clinical measures.
-
End with a short paragraph connecting these findings to intervention design (e.g., implications for bimanual or task-specific therapy).
Limitations
Consider adding:
-
The relatively small and possibly unbalanced sample (e.g., sex, severity).
-
The possibility that IdFAI (or equivalent constructs) might not capture qualitative aspects of performance.
Conclusion
You might strengthen the clinical message, for example: “These findings highlight the importance of integrating real-world activity measures such as accelerometry with clinical assessments to better guide individualised therapy planning in children with unilateral CP.”
Comments on the Quality of English LanguageThe manuscript is written in clear and generally well-structured English. However, minor language and style revisions would improve clarity and flow, particularly in the Introduction, Methods, and Discussion sections. Some sentences are long or overly detailed and could be simplified for better readability. A light professional proofreading is recommended to enhance consistency and precision in scientific terminology. Overall, the English quality is good, and the meaning is clearly conveyed.
Author Response
Section: Sensors Development
Manuscript ID: sensors-3934547
Manuscript title: Upper limb performance and capacity in children with unilateral cerebral palsy and their association with leisure participation
Dear Dr Thea Fang,
We would like to thank you for the opportunity to revise our manuscript, entitled " Upper limb performance and capacity in children with unilateral cerebral palsy and their association with leisure participation". We appreciate that you as well as the reviewers took the time to provide insightful comments et suggestions that have improved our paper. The changes are highlighted within the manuscript. Please also see below, in blue, a point-by-point response to the reviewers’ comments and concerns. All page numbers refer to the revised manuscript file with tracked changes. Please note that we have renumbered the comments from each reviewer to facilitate the reading of these comments and our responses.
All the reviewer’s comments were addressed while highlighting the required changes within the manuscript (in yellow). A point-by-point response (outlined in italics and blue) was provided to each reviewer’s comments followed, if applicable, by the extracts from the revised manuscript file with tracked changes (outlined in red and italics) as well as their location in the file (using page and line numbers).
Reviewer(s)' Comments to Author: Reviewer 1
1. This is a well-executed and timely study exploring how upper-limb performance and capacity relate to leisure participation in children with unilateral cerebral palsy (uCP). The topic is clinically meaningful and fits nicely within Sensors’interest in applying technology to rehabilitation contexts. The study design is solid, the methods are clearly explained, and the discussion connects well with the broader literature. That said, a few sections could be tightened to improve clarity and flow, mainly in the introduction and discussion. Adding a little more detail about the accelerometer procedures and emphasising the clinical relevance of the findings would make the paper even stronger.
Authors’ Response
We thank the reviewer for their positive feedback and constructive suggestions. We have improved the clarity and flow of the introduction and discussion, added details on accelerometer procedures to enhance methodological transparency, and strengthened the discussion of the clinical relevance of our findings.
Abstract
2. Include key numerical findings (e.g., correlation coefficients or effect sizes) to make the results more tangible.
Authors’ Response
We have added key numerical values (correlation coefficients and effect sizes) (page 1, line 36-37).
3. In the conclusion, use slightly softer wording, e.g., “no significant association was found” instead of “no association”, to reflect the exploratory nature of the analysis.
Authors’ Response
We have adjusted the wording in the conclusion to “the lack of significant association” to better reflect the exploratory nature of the analysis (page 1, line 37-38).
Introduction
4. The introduction could be shortened slightly by merging overlapping paragraphs around lines 70–100.
Authors’ Response
We have revised this section to improve conciseness and flow by merging the overlapping paragraphs. Specifically, we condensed the description of participation challenges and upper limb impairments in uCP. (page 3, line 91-104).
‘’Children with CP often encounter reduced participation in daily and leisure activities, which are essential for their development and social inclusion[6–8]. Compared to typically developing (TD) peers, they participate in fewer and less vigorous activities, often choosing passive, home-based pursuits [9–12]. Participation challenges in children with CP result from the complex combination of personal (e.g., capacity, performance) and environmental factors that can either facilitate or hinder their involvement [12,13]. While upper limb (UL) capacity and performance are crucial for daily and leisure activities such as self-care, play, and structured physical activities, their impact on participation in children with CP is underexplored [13]. Unilateral cerebral palsy (uCP), the most common CP subtype (about 44% of cases) [14], is a particularly relevant condition for examining the relationships between UL capacity, UL performance, and participation. Because uCP is characterized by lateralized sensorimotor impairments, it results in marked UL asymmetry and compensatory overuse of the dominant UL (i.e., the less affected limb) [15–17].’’
5. It would help to explain whythis link between upper-limb use and participation hasn’t been widely examined before (for instance, due to limitations of clinical tools or the variability of leisure activities).
Authors’ Response
We have revised the paragraph accordingly. Specifically, we now highlight that, to our knowledge, this relationship has not yet been investigated, partly due to the relatively recent adoption of accelerometry for assessing real-world upper limb performance, and partly because leisure activities are highly variable and context-dependent, making standardized assessment challenging. These additions can be found on page 3, lines 124–134.
‘’Previous research has examined the relationship between UL capacity and UL performance using accelerometry [22,23], or between UL capacity and participation [25–27], but none addressed all three dimensions concurrently. Moreover, the specific relationship between UL performance and leisure participation has not yet been investigated. This lack of research may be partly explained by the relatively recent adoption of accelerometry for assessing real-world UL performance [20] and the inherently variable and context-dependent nature of leisure activities, making their standardized assessment more challenging [28].’’
6. Consider adding one or two recent accelerometry studies for context (e.g., Beani et al., 2025; Coker-Bolt et al., 2016).
Authors’ Response
We have incorporated the two studies you recommended to provide additional context on the use of accelerometry. Specifically, we now refer to recent work demonstrating the feasibility and reliability of accelerometry for quantifying upper limb use both during intensive intervention contexts and in everyday life (21: Beani et al., 2025; 22: Coker-Bolt et al., 2016) (page 3, lines 113–115). In addition, our original manuscript already included a recent study examining associations between clinical assessments and accelerometry-derived metrics in daily life (Hedberg-Graff et al., 2025) (page 3, lines 115–119).
‘’It has been demonstrated that accelerometers can be a feasible and reliable method to quantify both more and less affected ULs use in children with uCP, whether during intensive intervention contexts or in everyday life [21,22].’’
Methods
7. Provide a short explanation of how you handled calibration between the two accelerometer models.
Authors’ Response
As requested, we have added a brief explanation in the Methods section describing how calibration between the two accelerometer models was handled (pages 5-6, lines 234–246).
‘‘To guarantee comparability between the two accelerometer models used in this study (ActiGraph GT9X Link and Axivity AX3), a pilot validation was carried out in our laboratory. Seven participants wore both devices simultaneously on the same wrist over a 2-day recording period during daily activities. To ensure methodological consistency, all data were analyzed using the same custom MATLAB scripts that process raw acceleration signals in the same way regardless of the device. Excellent agreement between ActiGraph and Axivity outputs was indicated by the ICC (single measures, consistency) of 0.981 (95% CI [0.893–0.997], p < 0.001). With a mean bias of 181.4 AC/min (SD = 186.3) and 95% limits of agreement ranging from -183.7 to 546.6 AC/min, Bland-Altman analysis further validated the good agreement. These results confirmed that both devices offer comparable measurements of UL activity under the same circumstances, which is in line with an earlier comparison between these two devices by Buchan et al. (2022) [31].’’
8. The shorter recording period for the uCP group (2 days vs 7 days) should be justified earlier in the Methods, not only in the Limitations.
Authors’ Response
The manuscript might not have been clear enough about the fact that all between-group comparisons were performed using a 2-day weekend monitoring. The other monitoring periods in TD kids were used only to test whether a 2-day monitoring is representative of a longer monitoring period. We acknowledge that a 2-day monitoring period might be noisier because of factors such as variability across weekends, weather, etc. However, it did not introduced a systematic bias across groups given that the same time window was used for all participants in our statistical analyses. This has now been clarified (Page 8, lines 339-342). In addition, the proportion of accelerometer models used did not differ significantly between groups (93.3% of TD participants wore the ActiGraph compared to 76.2% in the uCP group (p = 0.11)) and no significant difference in mean wear time was observed between groups (p = 0.284) (Page 8, lines 354-358). Therefore, neither adherence nor device model introduced any systematic bias between groups.
‘’All between-group comparisons were based on the 2-day weekend monitoring data. The longer monitoring periods (5 weekdays and 7 days) available for the TD group were used solely for secondary analyses assessing the reproducibility of weekend measurements relative to longer durations.’’
‘’The mean wear time for the TD group was 10.2 ± 2.76 hours per day. No significant difference in mean wear time was observed between groups (p = 0.284). Most TD participants (93.3%) wore the ActiGraph device. The distribution of accelerometer models (ActiGraph vs. Axivity) between groups did not differ significantly (p = 0.11).’’
9. Clarify whether a single examiner conducted all the JTHFT tests to control for rater variability.
Authors’ Response
The JTHFT was administered by three experienced evaluators (two occupational therapists and one physiotherapist). Since the uCP data were extracted from a larger longitudinal study involving multiple assessment time points and that has been ongoing for several years, it was not feasible to have a single examiner perform all evaluations. However, all evaluators received joint training and followed standardized JTHFT administration procedures to ensure consistency and minimize inter-rater variability. This clarification has been added to the Methods section (Page 6, lines 278–281).
‘’JTHFT was administered by three experienced evaluators (two occupational therapists and one physiotherapist). All evaluators received training and followed the same standardized administration procedures to ensure methodological consistency and minimize inter-rater variability across sessions.’’
Results
10. Simplify presentation by moving some detailed numeric results to supplementary material.
Authors’ Response
The section on reproducibility has been simplified by summarizing detailed numerical results in a supplementary table (Table A1 and A2).
11. Add a correlation summary table so that key relationships are easier to see at a glance.
Authors’ Response
A correlation summary table (Table 2) has been added to present all key relationships between upper limb capacity, performance, and participation.
12. Ensure all figure legends define abbreviations (e.g., D = dominant, ND = non-dominant)
Authors’ Response
We have reviewed and corrected all figure legends to ensure that abbreviations (e.g., D = dominant, ND = non-dominant) are clearly defined.
Discussion
13. Strengthen the mechanistic explanation. Why might no association have been found? Consider discussing compensatory strategies, cortical reorganisation, or task-specific motor control.
Authors’ Response
We appreciate the reviewer’s suggestion to strengthen the mechanistic explanation. In response, we have added a paragraph in the Discussion section addressing cortical reorganisation following early brain injury in children with uCP (page 13, lines 508–518).
‘’Another possible explanation involves cortical reorganisation following early brain lesions in uCP [56,57]. Due to the high plastic potential of the developing brain, motor control of the more affected UL can be reorganised either within the contralesional (unaffected) hemisphere or through spared regions of the ipsilesional (affected) hemisphere [56,57]. This adaptive reorganisation enables children with uCP to perform movements via alternative neural pathways [57,58]. However, these compensatory mechanisms often favor functional efficiency over precise motor control, which may result in relatively preserved gross motor abilities but persistent impairments in fine motor skills [57,58]. Such neuroplastic changes could therefore mask the relationship between capacities assessed through clinical tests and real-world performance as well as leisure participation.’’
14. Expand on the ecological validity of accelerometry compared with clinical measures.
Authors’ Response
The Recommendations section has been expanded to discuss in greater depth the ecological validity of accelerometry compared with traditional clinical assessments (pages 14, lines 584–602).
‘’Accelerometry offers an ecologically valid method to monitor how children use their ULs in their natural environment rather than during isolated tasks during clinical tests [69]. By continuously recording activity over multiple days, this approach provides an enhanced representation of motor behavior, free from examiner bias [70]. This is particularly relevant for populations living with hemiparesis, such as children with uCP, for whom accelerometers are particularly useful in detecting asymmetries in limb use and adaptive strategies that may remain unnoticed during short, structured assessments [69,71]. Although accelerometry offers important insights into the amount and symmetry of UL use, it does not assess qualitative aspects of movement [72,73]. This drawback points out the necessity for additional technologies capable of capturing fine motor skills in real-world contexts. Advancements in sensor fusion and multimodal motor assessment further expand the potential to overcome accelerometry’s inherent limitations [69]. Merging accelerometers with other sensing technologies, such as gyroscopes, markerless systems, and pressure or force sensors, allows assessment of both the kinematic and kinetic features of UL movement [69]. Sensor-integrated gloves show a potential innovation that provides a thorough evaluation of hand use by accurately monitoring finger movements and object manipulation [74]. The combination of these complementary technologies into unified sensor frameworks has a greater potential for the ecological validity and clinical sensitivity of UL assessments in children with uCP.’’
15. End with a short paragraph connecting these findings to intervention design (e.g., implications for bimanual or task-specific therapy).
Authors’ Response
A short paragraph has been added at the end of the Recommendations section to link our findings to clinical applications (page 14-15, lines 602–607).
‘’From a therapeutic perspective, integrating real-world activity measures into rehabilitation programs may advance individualized care by identifying distinct patterns of limb use [47,48]. Tools such as accelerometry and multimodal sensors allow clinicians to identify underused limbs and modify treatment strategies, including bimanual intensive therapy or devices providing feedback regarding limb use, to promote more balanced and efficient UL involvement in everyday situations [69,75].’’
Limitations
16. Consider adding: The relatively small and possibly unbalanced sample (e.g., sex, severity).
Authors’ Response
The limitations section has been expanded to acknowledge the relatively small and potentially unbalanced sample, including possible sex and severity differences (page 14, lines 577–578).
‘’The relatively small and potentially unbalanced sample in terms of sex distribution and severity levels may have influenced the generalizability of our findings.’’
17. Consider adding: The possibility that IdFAI (or equivalent constructs) might not capture qualitative aspects of performance.
Authors’ Response
We do not understand this comment given that the IdFAI was not used in the present study (assuming that the Reviewer is referring to “Identification of Functional Ankle Instability”).
Conclusion
18. You might strengthen the clinical message, for example: “These findings highlight the importance of integrating real-world activity measures such as accelerometry with clinical assessments to better guide individualised therapy planning in children with unilateral CP.”
Authors’ Response
We have revised the conclusion to include a clear clinical message emphasizing the integration of real-world activity measures such as accelerometry with standard clinical assessments (page 15, lines 623-627).
‘’Applying these methodological advancements is essential for clinical use, as combining real-world activity assessments, such as accelerometry and sensor-integrated gloves, with standard clinical measures can more effectively guide personalized therapy and detect significant changes in daily UL function in children with uCP.’’
19. Comments on the Quality of English Language
The manuscript is written in clear and generally well-structured English. However, minor language and style revisions would improve clarity and flow, particularly in the Introduction, Methods, and Discussion sections. Some sentences are long or overly detailed and could be simplified for better readability. A light professional proofreading is recommended to enhance consistency and precision in scientific terminology. Overall, the English quality is good, and the meaning is clearly conveyed.
Authors’ Response
We thank the reviewer for the positive feedback. The manuscript has been carefully revised to improve clarity, flow, and overall language quality.

Reviewer 2 Report
Comments and Suggestions for Authors
Dear Authors,
The study addresses an important question but fails to demonstrate sufficient methodological or conceptual novelty. The work’s main limitation is that it confirms well-known asymmetries in uCP without offering deeper mechanistic or technological insight. Furthermore, the mixed data acquisition design (weekend-only vs. full week) severely limits the reliability of between-group comparisons.
Below are detailed, critical comments intended to strengthen the rigor and interpretability of the work.
Introduction
-
The introduction is lengthy and overly descriptive. It summarizes prior literature adequately but does not sufficiently articulate the knowledge gap the study aims to fill.
-
The statement “the interrelationships between UL capacity, UL performance, and participation remain unexplored” is inaccurate — prior work (e.g., Bjornson et al., Arch Phys Med Rehabil, 2013; Hedberg-Graff et al., Disabil Rehabil, 2025) has addressed these associations.
-
The research hypothesis should be clearly stated and testable. Currently, objectives are descriptive, not inferential.
Methods
-
The cross-sectional design is acceptable, but the use of two distinct data collection frameworks (2-day vs. 7-day) introduces a major systematic bias. Although the authors try to justify this through a validation in TD children, the reasoning remains weak: the groups differ in monitoring context, adherence, and device models.
-
The validation between accelerometer models (ActiGraph GT9X vs. Axivity AX3) is insufficiently detailed. The statement “no significant differences were observed in the raw signals” is vague. Include quantitative metrics (correlation coefficients, Bland–Altman limits, bias values).
-
The use of a short monitoring period (2 days) in the clinical group is problematic: variability across weekends, weather, and daily schedules is not accounted for.
-
No a priori power analysis for correlation analyses is reported. With n=21, the study is underpowered to detect moderate associations (r ≈ 0.4).
-
Statistical assumptions (ANCOVA normality, homoscedasticity) are mentioned but not evidenced.
Results
-
The results confirm expected asymmetries in UL activity but do not contribute new understanding.
-
Reporting should include confidence intervals and effect sizes for all main outcomes.
-
The section “no significant correlations” is repeated for several comparisons — these could be summarized in a concise table rather than occupying multiple paragraphs.
-
Figures lack clarity: in Figure 1, labeling is crowded, and the color palette (red/blue) is non–colorblind-friendly.
Discussion
-
The discussion is overstated given the modest data. Statements such as “children adopt effective compensatory mechanisms” are speculative, as compensatory mechanisms were not directly measured.
-
There is excessive repetition of known findings (asymmetry, developmental disregard, contextual effects) without integration of new mechanistic insight or model development.
-
The discussion would benefit from engaging more deeply with recent literature on sensor fusion, multi-modal motor assessment, and ecological validity of accelerometry, which are core to Sensors.
-
The limitations section is transparent but too lenient—the two-day recording and dual-sensor methodology compromise external validity more severely than stated.
Conclusions
-
The conclusions are accurate but add little beyond prior knowledge. The final paragraph reads more like a summary of the abstract than a critical synthesis of implications.
-
The suggestion to use “sensor-integrated gloves” is interesting but peripheral to the current dataset.
Presentation and style
-
English language is mostly clear but verbose.
-
Several typos and formatting inconsistencies persist (e.g., “the child actually do” → “does”).
-
Redundant citations: [23] and [29] are identical; ensure reference list accuracy.
Summary of major concerns
-
Methodological inconsistency between uCP and TD data collection.
-
Insufficient validation between accelerometer models.
-
Low sample size and weak statistical power for correlational analyses.
-
Lack of novelty and overinterpretation of well-established findings.
-
Limited alignment with Sensors’ technological focus (i.e., the work uses off-the-shelf accelerometry without novel analytics).
Kind regards :)
Author Response
Section: Sensors Development
Manuscript ID: sensors-3934547
Manuscript title: Upper limb performance and capacity in children with unilateral cerebral palsy and their association with leisure participation
Dear Dr Thea Fang,
We would like to thank you for the opportunity to revise our manuscript, entitled " Upper limb performance and capacity in children with unilateral cerebral palsy and their association with leisure participation". We appreciate that you as well as the reviewers took the time to provide insightful comments et suggestions that have improved our paper. The changes are highlighted within the manuscript. Please also see below, in blue, a point-by-point response to the reviewers’ comments and concerns. All page numbers refer to the revised manuscript file with tracked changes. Please note that we have renumbered the comments from each reviewer to facilitate the reading of these comments and our responses.
All the reviewer’s comments were addressed while highlighting the required changes within the manuscript (in yellow). A point-by-point response (outlined in italics and blue) was provided to each reviewer’s comments followed, if applicable, by the extracts from the revised manuscript file with tracked changes (outlined in red and italics) as well as their location in the file (using page and line numbers).
Reviewer(s)' Comments to Author: Reviewer 2
1. The study addresses an important question but fails to demonstrate sufficient methodological or conceptual novelty. The work’s main limitation is that it confirms well-known asymmetries in uCP without offering deeper mechanistic or technological insight. Furthermore, the mixed data acquisition design (weekend-only vs. full week) severely limits the reliability of between-group comparisons.
Below are detailed, critical comments intended to strengthen the rigor and interpretability of the work.
Authors’ Response
We thank the reviewer for the constructive and valuable comments. We acknowledge the concerns raised regarding the study's originality and the heterogeneity in data acquisition. In response, we have strengthened the revised manuscript by clarifying the conceptual contribution of our work, focusing on the simultaneous assessment of upper limb capacity, real-world performance, and leisure participation. This area remains insufficiently explored in children with unilateral cerebral palsy. We have also expanded the methodological justification and discussion regarding the weekend-only data collection in the uCP group, highlighting our validation analyses within the TD group that support the reliability of this approach. Importantly, all between-group comparisons were performed using the weekend only data in both groups, to avoid bias in this comparison. This has been clarified in the manuscript (Page 8, lines 339-342). We believe these revisions strengthen the study's scientific contribution and methodological rigor.
‘’All between-group comparisons were based on the 2-day weekend monitoring data. The longer monitoring periods (5 weekdays and 7 days) available for the TD group were used solely for secondary analyses assessing the reproducibility of weekend measurements relative to longer durations.’’
Introduction
2. The introduction is lengthy and overly descriptive. It summarizes prior literature adequately but does not sufficiently articulate the knowledge gapthe study aims to fill. The statement “the interrelationships between UL capacity, UL performance, and participation remain unexplored” is inaccurate — prior work (e.g., Bjornson et al., Arch Phys Med Rehabil, 2013; Hedberg-Graff et al., Disabil Rehabil, 2025) has addressed these associations
Authors’ Response
We streamlined the introduction in the revised manuscript to highlight the conceptual framework and research gap. We now clarify that prior studies have explored these relationships either at a gross motor level (e.g., Bjornson et al., 2013 — to our knowledge, it is the only study that examined all three dimensions concurrently, though not focused on upper limb function) or by investigating specific associations such as between UL capacity and performance (e.g., Hedberg-Graff et al., 2025). To date, however, no study has simultaneously examined UL capacity, real-world performance (via accelerometry), and leisure participation in children with unilateral cerebral palsy, which is the specific focus of our work (Page 3, lines 124–131 and 136–138).
‘’Previous research has examined the relationship between UL capacity and UL performance using accelerometry [22,23], or between UL capacity and participation [25–27], but none addressed all three dimensions concurrently. Moreover, the specific relationship between UL performance and leisure participation has not yet been investigated. This lack of research may be partly explained by the relatively recent adoption of accelerometry for assessing real-world UL performance [20] and the inherently variable and context-dependent nature of leisure activities, making their standardized assessment more challenging [28].’’
‘’Despite these advances, no study has simultaneously examined UL capacity, real-world performance, and leisure participation specifically in children with uCP.’’
3. The research hypothesis should be clearly stated and testable. Currently, objectives are descriptive, not inferential.
Authors’ Response
We have reformulated the study objectives to include explicit and testable hypotheses aligned with the main research aims (Page 4, lines 158-180).
''This study had two main (clinical) objectives:
- To compare UL performance, measured by the intensity of UL activity, between children with uCP and TD children and between sides (dominant and non-dominant). We hypothesized that children with uCP would show reduced overall UL activity for both ULs, as well as a greater interlimb asymmetry compared to TD children.
- To explore the associations between UL performance, UL capacity, and their participation in leisure activities outside the school setting in children with uCP. We hypothesized that higher UL capacity and UL performance would be associated with higher leisure participation.
Two secondary (methodological) objectives were defined in relation to the main objectives:
- To examine whether a 2-day (weekend) measurement period provides a reliable estimate of UL performance compared to 5-day (weekdays) and 7-day (full week) periods, and to evaluate the reproducibility of children’s activity patterns across these different recording durations. This validation was only performed within the TD group, because the data for the uCP participants were limited to the weekend recordings. This limitation was explained by their participation in a separate longitudinal study with multiple evaluations of the effects of bimanual therapy, where the use of accelerometry was limited to weekends to ease the burden on both children and their families before, during, and after the intervention.
- To assess the effect of age on the UL performance, given the large heterogeneity in our group in terms of age.''
Methods
4. The cross-sectional design is acceptable, but the use of two distinct data collection frameworks (2-day vs. 7-day) introduces a major systematic bias. Although the authors try to justify this through a validation in TD children, the reasoning remains weak: the groups differ in monitoring context, adherence, and device models.
Authors’ Response
The manuscript might not have been clear enough about the fact that all between-group comparisons were performed using a 2-day weekend monitoring. The other monitoring periods in TD kids were used only to test whether a 2-day monitoring is representative of a longer monitoring period. We acknowledge that a 2-day monitoring period might be noisier because of factors such as variability across weekends, weather, etc. However, it did not introduced a systematic bias across groups given that the same time window was used for all participants in our statistical analyses. This has now been clarified (Page 8, lines 339-342). In addition, the proportion of accelerometer models used did not differ significantly between groups (93.3% of TD participants wore the ActiGraph compared to 76.2% in the uCP group (p = 0.11)) and no significant difference in mean wear time was observed between groups (p = 0.284) (Page 8, lines 354-358). Therefore, neither adherence nor device model introduced any systematic bias between groups.
‘’All between-group comparisons were based on the 2-day weekend monitoring data. The longer monitoring periods (5 weekdays and 7 days) available for the TD group were used solely for secondary analyses assessing the reproducibility of weekend measurements relative to longer durations.’’
‘’The mean wear time for the TD group was 10.2 ± 2.76 hours per day. No significant difference in mean wear time was observed between groups (p = 0.284). Most TD par-ticipants (93.3%) wore the ActiGraph device. The distribution of accelerometer models (ActiGraph vs. Axivity) between groups did not differ significantly (p = 0.11).’’
5. The validation between accelerometer models (ActiGraph GT9X vs. Axivity AX3) is insufficiently detailed. The statement “no significant differences were observed in the raw signals” is vague. Include quantitative metrics (correlation coefficients, Bland–Altman limits, bias values).
Authors’ Response
We have revised the manuscript to include detailed quantitative results of the inter-device validation (Page5-6, lines 234-246).
‘’To guarantee comparability between the two accelerometer models used in this study (ActiGraph GT9X Link and Axivity AX3), a pilot validation was carried out in our laboratory. Seven participants wore both devices simultaneously on the same wrist over a 2-day recording period during daily activities. To ensure methodological consistency, all data were analyzed using the same custom MATLAB scripts that process raw acceleration signals in the same way regardless of the device. Excellent agreement between ActiGraph and Axivity outputs was indicated by the ICC (single measures, consistency) of 0.981 (95% CI [0.893–0.997], p < 0.001). With a mean bias of 181.4 AC/min (SD = 186.3) and 95% limits of agreement ranging from -183.7 to 546.6 AC/min, Bland-Altman analysis further validated the good agreement. These results confirmed that both devices offer comparable measurements of UL activity under the same circumstances, which is in line with an earlier comparison between these two devices by Buchan et al. (2022) [31].’’
6. The use of a short monitoring period (2 days) in the clinical group is problematic: variability across weekends, weather, and daily schedules is not accounted for.
Authors’ Response
We agree with the reviewer that short-term monitoring may be affected by contextual variability (e.g., weather, weekend routines, and family schedules). This point has now been explicitly acknowledged in the Limitations section (Page14, lines 559–562). Nevertheless, previous research has demonstrated that short monitoring periods can provide reliable estimates of motor performance and physical activity in children with CP. For example, Gerber et al. (Ann Phys Rehabil Med, 2021) showed that two consecutive days of sensor-based measurements are reliable for this population's walking performance and physical activity. We have added this reference and clarification to the Methods section (2.3 Study protocol; page 5, lines 215–218) to explicitly justify the use of a shorter recording period. Together with our validation analyses in TD children, this supports the methodological soundness of using a 2-day protocol in the uCP group.
‘'Furthermore, external factors such as weather conditions, weekend routines, or family schedules may also have influenced activity levels during the weekend. This could have introduced additional variability that was not fully controlled.’’
‘’Moreover, previous research has shown that short monitoring periods of two consecutive days can yield reliable estimates of daily physical activity and gait performance in children with CP, with intraclass correlation coefficients (ICC) ranging from 0.70 to 0.98 when using wearable inertial sensors [30].’’
7. No a priori power analysis for correlation analyses is reported. With n=21, the study is underpowered to detect moderate associations (r ≈ 0.4).
Authors’ Response
We acknowledge the reviewer’s point regarding statistical power. The relatively small sample size (n = 21) reflects the difficulty of recruiting children with uCP, a population with specific inclusion criteria. We now highlight this limitation in the manuscript (Page 14: lines 574–577).
‘’Another limitation concerns the relatively small sample size in the uCP group (n = 21), which reflects the well-known challenges of recruiting this clinical population. This limited number of participants may have reduced the statistical power to detect moderate correlations.’’
8. Statistical assumptions (ANCOVA normality, homoscedasticity) are mentioned but not evidenced.
Authors’ Response
We have now included the relevant test statistics in the Methods section to document that all ANCOVA assumptions were verified (Page 7, lines 308–315).
‘’Assumptions of normality (TD group: Shapiro–Wilk W = 0.941, p = 0.096 for the dominant side; W = 0.958, p = 0.271 for the non-dominant side; group with uCP: Shapiro–Wilk W = 0.968, p = 0.697 for the dominant side; W = 0.949, p = 0.333 for the non-dominant side), homogeneity of variance (Levene’s test: F = 0.029, p = 0.866 for the dominant side and F = 0.962, p = 0.332 for the non-dominant side), homogeneity of regression slopes (Group × Age interaction: F = 0.4, p = 0.53 for the dominant side; F = 1.313, p = 0.258 for the non-dominant side), and linearity between the dependent variable and the covariate were verified before conducting the ANCOVA. ‘’
Results
9. The results confirm expected asymmetries in UL activity but do not contribute new understanding.
Authors’ Response
We acknowledge the reviewer’s comment. While the asymmetry in upper limb activity aligns with previous evidence, our study provides additional value by quantifying this asymmetry in real-world contexts using accelerometry and examining its relationship with upper limb capacity and participation. Moreover, it offers novel insight by showing that dominant limb performance does not differ from that of typically developing peers. These results suggest the presence of compensatory behaviors that help maintain overall activity levels despite motor asymmetry.
10. Reporting should include confidence intervals and effect sizes for all main outcomes.
Authors’ Response
Confidence intervals (95 %) have now been added for all main comparisons, as requested (Page 9, lines 373-379).
‘’Multiple comparison analyses revealed a significant difference between groups (group with uCP vs. TD group) for the non-dominant side (95 % CI [-2150;-1278], p < 0.001, ηp2 = 0.565). However, no significant difference was observed between groups for the dominant side (95 % CI [-597, 491], p = 0.846, ηp2 =0.001). A significant difference between sides (non-dominant vs. dominant) was found for the uCP group (95 % CI [-2120, -1563], p < 0.001, ηp2 = 0.786), while no such difference was found for the TD group (95 % CI [-413, 52], p = 0.125, ηp2 =0.048).’’
11. The section “no significant correlations” is repeated for several comparisons — these could be summarized in a concise table rather than occupying multiple paragraphs.
Authors’ Response
We have revised the section to remove repeated text and now summarize all correlation results in a single table (Table 2).
11. Figures lack clarity: in Figure 1, labeling is crowded, and the color palette (red/blue) is non–colorblind-friendly.
Authors’ Response
Figure 1 has been revised to improve labeling and updated with a colorblind-friendly, as suggested by the GraphPad Prism software.
Discussion
12. The discussion is overstated given the modest data. Statements such as “children adopt effective compensatory mechanisms” are speculative, as compensatory mechanisms were not directly measured.
Authors’ Response
We agree that compensatory mechanisms were not directly assessed in the present study. Accordingly, we have revised the discussion to adopt a more cautious interpretation of our findings. We explicitly acknowledge that this interpretation remains speculative and is based on consistency with previous literature rather than direct measurement (Page 12, lines 472-474)
‘’Such an interpretation, while remaining speculative, consistent with previous evidence indicating that children with uCP may develop adaptive mechanisms to sustain overall activity despite unilateral limitations [21,24,42,43].’’
13. There is excessive repetition of known findings (asymmetry, developmental disregard, contextual effects) without integration of new mechanistic insight or model development.
Authors’ Response
We have revised the discussion to better integrate new mechanistic perspectives. Specifically, we added a paragraph discussing the potential contribution of cortical reorganisation and neuroplastic adaptation in explaining the absence of associations between upper limb capacity, performance, and participation in children with uCP. This addition provides a broader mechanistic framework for interpreting our findings (Page 13, lines 508-518).
“Another possible explanation involves cortical reorganisation following early brain lesions in uCP [56,57]. Due to the high plastic potential of the developing brain, motor control of the more affected UL can be reorganised either within the contralesional (unaffected) hemisphere or through spared regions of the ipsilesional (affected) hemisphere [56,57]. This adaptive reorganisation enables children with uCP to perform movements via alternative neural pathways [57,58]. However, these compensatory mechanisms often favor functional efficiency over precise motor control, which may result in relatively preserved gross motor abilities but persistent impairments in fine motor skills [57,58]. Such neuroplastic changes could therefore mask the relationship between capacities assessed through clinical tests and real-world performance as well as leisure participation.’’
14. The discussion would benefit from engaging more deeply with recent literature on sensor fusion, multi-modal motor assessment, and ecological validity of accelerometry, which are core to Sensors.
Authors’ Response
We have expanded the discussion to incorporate recent literature on sensor fusion, multi-modal motor assessment, and the ecological validity of accelerometry. This addition highlights how integrating accelerometry with complementary sensing modalities (e.g., gyroscopes, force sensors) can enhance the ecological and functional relevance of upper limb assessments in children with uCP (Page 14, lines 595-602).
“Merging accelerometers with other sensing technologies, such as gyroscopes, markerless systems, and pressure or force sensors, allows assessment of both the kinematic and kinetic features of UL movement [69]. Sensor-integrated gloves show a potential innovation that provides a thorough evaluation of hand use by accurately monitoring finger movements and object manipulation [74]. The combination of these complementary technologies into unified sensor frameworks has a greater potential for the ecological validity and clinical sensitivity of UL assessments in children with uCP. “
15. The limitations section is transparent but too lenient—the two-day recording and dual-sensor methodology compromise external validity more severely than stated.
Authors’ Response
We acknowledge the reviewer’s concern; however, our secondary analyses showed good reproducibility between short and longer monitoring periods, and excellent agreement between accelerometer models (ICC = 0.98; data that has been added to the revised version of our manuscript). These results indicate that the two-day recording and dual-sensor approach did not substantially compromise the validity of our findings.
Conclusion
16. The conclusions are accurate but add little beyond prior knowledge. The final paragraph reads more like a summary of the abstract than a critical synthesis of implications.
Authors’ Response
The conclusion has been substantially revised to move beyond a summary of results and to provide a more interpretative synthesis. The revised conclusion now focuses on the broader implications of our findings, emphasizing the complexity of participation (Page 15, lines 611-630).
‘’Conclusions
This study highlights the pronounced asymmetry in daily UL performance within children living with uCP, showing significantly reduced use of the non-dominant side. In contrast, the performance of the dominant side did not differ from that of TD peers, suggesting that compensatory patterns are intended to maintain overall activity levels regardless of motor impairments. Although these interpretations are hypothetical, compensatory mechanisms were not directly assessed in this study. Additionally, the lack of significant associations between UL capacity, UL performance, and leisure participation highlights the multifactorial nature of participation, which is determined by multiple factors beyond motor function. Nonetheless, accelerometry offers valuable features for objectively quantifying daily UL use, it mainly captures the amount and intensity of gross motor movements but does not offer information on qualitative aspects such as fine motor skills. These findings emphasize the need to complement accelerometry with additional tools capable of assessing both qualitative and quantitative dimensions of movement. Applying these methodological advancements is essential for clinical use, as combining real-world activity assessments, such as accelerometry and sensor-integrated gloves, with standard clinical measures can more effectively guide personalized therapy and detect significant changes in daily UL function in children with uCP. Our findings suggest that while accelerometry provides objective insights into daily upper limb activity, participation outcomes in children with uCP might depend more on contextual and psychosocial than on motor factors.”
17. The suggestion to use “sensor-integrated gloves” is interesting but peripheral to the current dataset.
Authors’ Response
The conclusion was revised to clarify that the mention of sensor-integrated gloves is a prospective recommendation, not directly derived from the present data (Page 15, lines 623-627).
“Applying these methodological advancements is essential for clinical use, as combining real-world activity assessments, such as accelerometry and sensor-integrated gloves, with standard clinical measures can more effectively guide personalized therapy and detect significant changes in daily UL function in children with uCP.”
18. Presentation and style
- English language is mostly clear but verbose.
- Several typos and formatting inconsistencies persist (e.g., “the child actually do” → “does”).
- Redundant citations: [23] and [29] are identical; ensure reference list accuracy.
Authors’ Response
The manuscript has been carefully edited to reduce verbosity and correct typos and formatting inconsistencies. The redundant citation ([23] and [29]) has been removed, and the reference list has been verified for accuracy.
19. Summary of major concerns
- Methodological inconsistency between uCP and TD data collection.
- Insufficient validation between accelerometer models.
- Low sample size and weak statistical power for correlational analyses.
- Lack of novelty and overinterpretation of well-established findings.
- Limited alignment with Sensors’ technological focus (i.e., the work uses off-the-shelf accelerometry without novel analytics).
Authors’ Response
As this is a summary, these points have been addressed in response to the detailed comments above.

Reviewer 3 Report
Comments and Suggestions for Authors
-
The first editorial remark: parentheses used for citations should be replaced with square brackets.
-
The introduction correctly refers to the three key ICF concepts (capacity, performance, participation); however, it lacks a deeper explanation of how these concepts specifically relate to the functioning of children with CP. For example, it could be pointed out that CP typically involves a discrepancy between “capacity” (what a child can do in clinical tests) and “performance” (what they actually do in their everyday environment) — one of the main research issues addressed in this study. Clarifying this relationship would provide a smoother transition to the following sections of the text.
-
uCP is introduced as the most common form of CP, but the transition from the general description of CP to uCP is too abrupt. A clear scientific justification is missing as to why uCP is considered the best model for analysing the relationships between capacity, performance, and participation. It could be emphasised that uCP — due to its asymmetrical deficits and the phenomenon of “developmental disregard” — is particularly useful for studying the discrepancy between ability and actual limb use.
-
The introduction refers to numerous studies on UL capacity, performance, and participation but does not clearly indicate the research gap this paper aims to address. It would be helpful to add a concise sentence highlighting that, despite many studies examining each of these components separately, their interrelationships — particularly in unilateral CP — remain poorly understood, forming the rationale for the present study.
-
The section describing the study objectives (lines 109–123) is very detailed and difficult to follow. It would be beneficial to separate them clearly into:
a. Main objectives (clinical hypotheses): concerning group differences and relationships between UL capacity, performance, and participation variables.
b. Secondary objectives (methodological): concerning measurement duration and age effects.
Such structuring would improve clarity and underline the significance of the main research questions in the context of existing literature. -
The authors describe the transition from ActiGraph GT9X to Axivity AX3, providing technical reasons and validation tests. However, detailed information on the validation results is missing — for example, the number of tested signals, correlation coefficients between devices, or references confirming measurement equivalence. It should be suggested to include a short description of the comparative test results (e.g. “Pearson’s r > 0.95, p < 0.001”), which would enhance methodological credibility and transparency.
-
The choice of a two-day recording period for the uCP group is presented as a practical compromise, yet there is no empirical justification that weekend measurement is representative of a child’s typical activity level. A reference to previous research (e.g. showing that two days ensure ≥80% measurement stability) should be added, or it should be mentioned that this issue was one of the secondary validation aims of the study. The current description may suggest limited comparability between groups.
-
The section describing signal filtering and the calculation of vector magnitude (AC) is technically correct but too brief for full replication. It would be helpful to specify:
a. the parameters of the band-pass filter (cut-off frequencies);
b. the method of distinguishing wear and non-wear periods (automatic algorithm vs visual inspection);
c. the criteria for defining a “valid day” (e.g. minimum of eight hours of wear time).
Such details are essential for transparency and reproducibility.
d. In the description of the JTHFT and CAPE, reliability and scoring ranges are provided, but it is not specified:
e. whether data were analysed as raw times/scores or normalised values (e.g. age-adjusted);
f. which CAPE dimensions were included in the correlation analyses (only “intensity” or other indices as well).
Clarifying this aspect would allow readers to understand precisely which variables entered the statistical analysis and how to interpret the correlations obtained. -
Line 231 – “Bonferroni correction applied to adjust for multiple tests.” – please indicate how the Bonferroni correction affected the p-values in the text.
-
Lines 231–234 – “Effect sizes were calculated using partial eta-squared (ηp²) and interpreted as small (0.01–0.059), medium (0.06–0.13), and large (≥0.14).” – there are various interval classifications for effect size; please provide a citation supporting these particular cut-off values.
-
Lines 237–239 – “Correlation coefficients were interpreted using standard benchmarks: small (r = 0.10–0.29), moderate (r = 0.30–0.49), and large (r ≥ 0.50)” – this is a substantive error. As mentioned, several sets of benchmarks exist. The authors applied a standard classification, but the values recommended for physiotherapy research should be used instead: small, medium, and large effect sizes should be considered 0.3, 0.5, and 0.6, respectively. Please refer to the guidelines in DOI: 10.1016/j.apmr.2025.05.013.
-
“Insert Table 1 around here” – according to MDPI guidelines, tables should appear within the text; please insert the table in the appropriate place.
-
The discussion accurately describes the asymmetry in limb use but focuses mainly on theoretical explanations (“developmental disregard”) without exploring the clinical and therapeutic implications. It would be worthwhile to indicate how this finding may influence rehabilitation strategies (e.g. the importance of bimanual therapy, CIMT, or motivational interventions aimed at increasing use of the more affected limb). The current interpretation remains too static and does not show how the data might inform clinical practice.
-
Although differences in measurement duration between groups are mentioned in the “Limitations” section, there is no discussion of how this might have affected the main conclusions regarding asymmetry and lack of correlation. In children with uCP, the shorter recording period (weekend only) could have reduced activity variability and limited the range of observed behaviours. This issue should be discussed within the interpretation of results, not solely in the limitations section — for instance, whether the correlations might have been weakened by this effect.
-
The authors rightly observe the lack of association between manual ability (JTHFT), performance (accelerometry), and participation (CAPE), but the interpretation is limited to differences in measurement types. It should be added that such a lack of correlation may also stem from non-linear or indirect relationships (e.g. mediated by psychosocial factors, perceived competence, or family support). A short note on the need for multivariate or mediation analyses (e.g. SEM, mediation analysis) would strengthen the scientific weight of this conclusion.
-
The recommendations regarding new technologies (e.g. sensor gloves) are interesting but not directly derived from the study data. It would be better first to relate them explicitly to a specific limitation of accelerometry (e.g. its inability to assess movement quality) and only then propose the technological alternative.
In addition, the “Conclusions” section lacks a clear statement on how the findings confirm or refute the research hypotheses and what they contribute to existing literature. Instead of a summary, it would be preferable to end with a stronger interpretative statement such as:“Our findings suggest that while accelerometry provides objective insights into daily upper limb activity, participation outcomes in children with uCP depend more on contextual and psychosocial than on motor factors.” – please consider this.
Author Response
Section: Sensors Development
Manuscript ID: sensors-3934547
Manuscript title: Upper limb performance and capacity in children with unilateral cerebral palsy and their association with leisure participation
Dear Dr Thea Fang,
We would like to thank you for the opportunity to revise our manuscript, entitled " Upper limb performance and capacity in children with unilateral cerebral palsy and their association with leisure participation". We appreciate that you as well as the reviewers took the time to provide insightful comments et suggestions that have improved our paper. The changes are highlighted within the manuscript. Please also see below, in blue, a point-by-point response to the reviewers’ comments and concerns. All page numbers refer to the revised manuscript file with tracked changes. Please note that we have renumbered the comments from each reviewer to facilitate the reading of these comments and our responses.
All the reviewer’s comments were addressed while highlighting the required changes within the manuscript (in yellow). A point-by-point response (outlined in italics and blue) was provided to each reviewer’s comments followed, if applicable, by the extracts from the revised manuscript file with tracked changes (outlined in red and italics) as well as their location in the file (using page and line numbers).
Reviewer(s)' Comments to Author: Reviewer 3
Introduction
1. The first editorial remark: parentheses used for citations should be replaced with square brackets.
Authors’ Response
All citation parentheses have been replaced with square brackets throughout the manuscript.
2. The introduction correctly refers to the three key ICF concepts (capacity, performance, participation); however, it lacks a deeper explanation of how these concepts specifically relate to the functioning of children with CP. For example, it could be pointed out that CP typically involves a discrepancy between “capacity” (what a child can do in clinical tests) and “performance” (what they actually do in their everyday environment) — one of the main research issues addressed in this study. Clarifying this relationship would provide a smoother transition to the following sections of the text.
Authors’ Response
We thank the reviewer for the insightful and valuable comments. We have added a short explanation of the well-documented discrepancy between capacity and performance in CP, emphasizing that children may demonstrate abilities in standardized assessments that are not reflected in their everyday arm use (Page2, lines 64–66).
‘’For children with CP, a discrepancy tends to exist between the capacity of the impaired limb and its performance [4,5]. This gap may have important implications for participation and represents a central issue addressed in the present study.”
3. uCP is introduced as the most common form of CP, but the transition from the general description of CP to uCP is too abrupt. A clear scientific justification is missing as to why uCP is considered the best model for analysing the relationships between capacity, performance, and participation. It could be emphasised that uCP — due to its asymmetrical deficits and the phenomenon of “developmental disregard” — is particularly useful for studying the discrepancy between ability and actual limb use.
Authors’ Response
We improved the transition from the general description of CP to uCP by introducing the concept of asymmetrical deficits at the beginning of the paragraph (Page3, lines 99-104).
’’ Unilateral cerebral palsy (uCP), the most common CP subtype (about 44% of cases) [14], is a particularly relevant condition for examining the relationships between UL capacity, UL performance, and participation. Because uCP is characterized by lateralized sensorimotor impairments, it results in marked UL asymmetry and compensatory overuse of the dominant UL (i.e., the less affected limb) [15–17].’’
4. The introduction refers to numerous studies on UL capacity, performance, and participation but does not clearly indicate the research gap this paper aims to address. It would be helpful to add a concise sentence highlighting that, despite many studies examining each of these components separately, their interrelationships — particularly in unilateral CP — remain poorly understood, forming the rationale for the present study.
Authors’ Response
We have clarified the research gap by explicitly stating that although many studies have examined UL capacity, performance, and participation separately, their interrelationships remain poorly understood, particularly in unilateral CP (page 3, lines 124–131 and lines 136-138).
’’ Previous research has examined the relationship between UL capacity and UL performance using accelerometry [22,23], or between UL capacity and participation [25–27], but none addressed all three dimensions concurrently. Moreover, the specific relationship between UL performance and leisure participation has not yet been investigated. This lack of research may be partly explained by the relatively recent adoption of accelerometry for assessing real-world UL performance [20] and the inherently variable and context-dependent nature of leisure activities, making their standardized assessment more challenging [28].”
’’Despite these advances, no study has simultaneously examined UL capacity, real-world performance, and leisure participation specifically in children with uCP.’’
5. The section describing the study objectives (lines 109–123) is very detailed and difficult to follow. It would be beneficial to separate them clearly into:
Main objectives (clinical hypotheses): concerning group differences and relationships between UL capacity, performance, and participation variables.
b. Secondary objectives (methodological): concerning measurement duration and age effects.
Such structuring would improve clarity and underline the significance of the main research questions in the context of existing literature.
Authors’ Response
We have restructured the objectives section by clearly distinguishing main (clinical) objectives from secondary (methodological) objectives (Page 4, lines 158-180)
"This study had two main (clinical) objectives:
- To compare UL performance, measured by the intensity of UL activity, between children with uCP and TD children and between sides (dominant and non-dominant). We hypothesized that children with uCP would show reduced overall UL activity for both ULs, as well as a greater interlimb asymmetry compared to TD children.
- To explore the associations between UL performance, UL capacity, and their participation in leisure activities outside the school setting in children with uCP. We hypothesized that higher UL capacity and UL performance would be associated with higher leisure participation.
Two secondary (methodological) objectives were defined in relation to the main objectives:
- To examine whether a 2-day (weekend) measurement period provides a reliable estimate of UL performance compared to 5-day (weekdays) and 7-day (full week) periods, and to evaluate the reproducibility of children’s activity patterns across these different recording durations. This validation was only performed within the TD group, because the data for the uCP participants were limited to the weekend recordings. This limitation was explained by their participation in a separate longitudinal study with multiple evaluations of the effects of bimanual therapy, where the use of accelerometry was limited to weekends to ease the burden on both children and their families before, during, and after the intervention.
- To assess the effect of age on the UL performance, given the large heterogeneity in our group in terms of age."
6. The authors describe the transition from ActiGraph GT9X to Axivity AX3, providing technical reasons and validation tests. However, detailed information on the validation results is missing — for example, the number of tested signals, correlation coefficients between devices, or references confirming measurement equivalence. It should be suggested to include a short description of the comparative test results (e.g. “Pearson’s r > 0.95, p < 0.001”), which would enhance methodological credibility and transparency.
Authors’ Response
We have expanded the methodological description to include details of the pilot validation protocol and quantitative metrics (Page 5-6, lines 234-245).
‘’To guarantee comparability between the two accelerometer models used in this study (ActiGraph GT9X Link and Axivity AX3), a pilot validation was carried out in our laboratory. Seven participants wore both devices simultaneously on the same wrist over a 2-day recording period during daily activities. To ensure methodological consistency, all data were analyzed using the same custom MATLAB scripts that process raw acceleration signals in the same way regardless of the device. Excellent agreement between ActiGraph and Axivity outputs was indicated by the ICC (single measures, consistency) of 0.981 (95% CI [0.893–0.997], p < 0.001). With a mean bias of 181.4 AC/min (SD = 186.3) and 95% limits of agreement ranging from -183.7 to 546.6 AC/min, Bland-Altman analysis further validated the good agreement. These results confirmed that both devices offer comparable measurements of UL activity under the same circumstances, which is in line with an earlier comparison between these two devices by Buchan et al. (2022) [31].”
7. The choice of a two-day recording period for the uCP group is presented as a practical compromise, yet there is no empirical justification that weekend measurement is representative of a child’s typical activity level. A reference to previous research (e.g. showing that two days ensure ≥80% measurement stability) should be added, or it should be mentioned that this issue was one of the secondary validation aims of the study. The current description may suggest limited comparability between groups.
Authors’ Response
The two-day monitoring period was predetermined by the design of the longitudinal study from which the uCP data were drawn. However, we refer to Gerber et al. (2019) to indicate that short monitoring durations can still provide reliable estimates of daily activity in children with CP, supporting the adequacy of this approach (Page 5, lines 215-218).
’’ Moreover, previous research has shown that short monitoring periods of two consecutive days can yield reliable estimates of daily physical activity and gait performance in children with CP, with intraclass correlation coefficients (ICC) ranging from 0.70 to 0.98 when using wearable inertial sensors [30].”
8. The section describing signal filtering and the calculation of vector magnitude (AC) is technically correct but too brief for full replication. It would be helpful to specify:
the parameters of the band-pass filter (cut-off frequencies);
Authors’ Response
The section has been revised to include the parameters of the band-pass filter (Page 6, lines 250-255).
’’ Following the approach described by Poitras et al. (2020), a continuous 8th-order bandpass filter was optimized using the developed algorithm in MATLAB to replicate the frequency response of the commercial activity count filter. The final continuous parameters were discretized at 100 Hz to match the sampling rate used in this study. This method ensures numerical stability and allows for accurate filtering of high-frequency motion data.”
9. the method of distinguishing wear and non-wear periods (automatic algorithm vs visual inspection);
Authors’ Response
The method used to identify non-wear periods was already specified in the manuscript. As stated, non-wear time and sleep were excluded through visual inspection and manual handling of the data to ensure accurate detection and consistency across participants (Page 6, lines 263-266).
10. the criteria for defining a “valid day” (e.g. minimum of eight hours of wear time).
Authors’ Response
We thank the reviewer for this helpful suggestion. Following this comment, the definition of a valid recording day has been specified in the Accelerometry data collection and processing section (Page 6, lines 267-269). A recording day was considered valid if the accelerometer was worn for a minimum average duration of 6 hours per day across the two weekend days, consistent with previous research in children with cerebral palsy using the ActiGraph (Hulst et al., 2023). Analyses were updated after excluding the four TD participants whose wear time was below this threshold. The final sample therefore included 30 TD and 21 uCP participants (Page 8, lines 347-358). Importantly, no significant difference in mean wear time was observed between groups (p = 0.284), indicating comparable adherence across groups. This adjustment did not affect the study conclusions.
’’ A recording day was defined as valid when the accelerometer was worn for a minimum average duration of 6 hours per day across the two weekend days. This criterion is consistent with previous research on children with cerebral palsy using ActiGraph [33].
“Four TD participants were excluded because their average wear time was below 6 hours per day across the two weekend days. The final sample, therefore, included 30 subjects in the TD group and 21 subjects in the uCP group. For the TD group, most of the participants were male (62.1%; not significantly different from uCP group, p=0.59), with a mean age of 10.6±2.11 years (not significantly different from uCP group, p=0.41), and 86.7% were right-handed. Dominance differed between groups (p=0.007, which was expected given the balanced number of uCP participants for whom the more affected side was the right side vs. the left side). The mean wear time for the TD group was 10.2 ± 2.76 hours per day. No significant difference in mean wear time was observed between groups (p = 0.284). Most TD participants (93.3%) wore the ActiGraph device. The distribution of accelerometer models (ActiGraph vs. Axivity) between groups did not differ significantly (p = 0.11).”
11. Such details are essential for transparency and reproducibility.
In the description of the JTHFT and CAPE, reliability and scoring ranges are provided, but it is not specified:
e. whether data were analysed as raw times/scores or normalised values (e.g. age-adjusted);
f. which CAPE dimensions were included in the correlation analyses (only “intensity” or other indices as well).
Authors’ Response
We have clarified in the revised manuscript that raw JTHFT completion times were used without age normalization, and only the overall CAPE intensity score was included in the correlation analyses. These details are now specified in the Statistical Analysis (Page 7, lines 325-328).
’’ For the JTHFT, raw completion times (in seconds) were used for each hand without age normalization. For the CAPE, only the overall intensity score was included in the correlation analyses, representing the intensity of participation across all activity types.”
12. Clarifying this aspect would allow readers to understand precisely which variables entered the statistical analysis and how to interpret the correlations obtained. Line 231 – “Bonferroni correction applied to adjust for multiple tests.” – please indicate how the Bonferroni correction affected the p-values in the text.
Authors’ Response
This has been clarified in the revised manuscript. The Statistical analyses section now specifies that Bonferroni corrections were applied to the p-values, and that the corrected p-values are reported in the text (Page 7, lines 318-320).
“Bonferroni corrections were applied to the p-values to adjust for multiple testing, and the corrected p-values are reported.”
13. Lines 231–234 – “Effect sizes were calculated using partial eta-squared (ηp²) and interpreted as small (0.01–0.059), medium (0.06–0.13), and large (≥0.14).” – there are various interval classifications for effect size; please provide a citation supporting these particular cut-off values.
Authors’ Response
A reference (39. Richardson JTE. Eta squared and partial eta squared as measures of effect size in educational research. Educ Res Rev. janv 2011;6(2):135‑47) has been added to support the interpretation thresholds for partial eta-squared (ηp²) values (Page 7, line 321).
14. Lines 237–239 – “Correlation coefficients were interpreted using standard benchmarks: small (r = 0.10–0.29), moderate (r = 0.30–0.49), and large (r ≥ 0.50)” – this is a substantive error. As mentioned, several sets of benchmarks exist. The authors applied a standard classification, but the values recommended for physiotherapy research should be used instead: small, medium, and large effect sizes should be considered 0.3, 0.5, and 0.6, respectively. Please refer to the guidelines in DOI: 10.1016/j.apmr.2025.05.013.
Authors’ Response
We have added the requested reference (DOI: 10.1016/j.apmr.2025.05.013). Correlation coefficients are now interpreted using the following thresholds: small (r ≥ 0.30), medium (r ≥ 0.50), and large (r ≥ 0.60) (Page 7, line 329).
15. “Insert Table 1 around here” – according to MDPI guidelines, tables should appear within the text; please insert the table in the appropriate place.
Authors’ Response
We have revised the manuscript to comply with MDPI formatting guidelines by inserting Table 1 and 2 directly within the text at the appropriate location.
16. The discussion accurately describes the asymmetry in limb use but focuses mainly on theoretical explanations (“developmental disregard”) without exploring the clinical and therapeutic implications. It would be worthwhile to indicate how this finding may influence rehabilitation strategies (e.g. the importance of bimanual therapy, CIMT, or motivational interventions aimed at increasing use of the more affected limb). The current interpretation remains too static and does not show how the data might inform clinical practice.
Authors’ Response
We have revised the discussion to include a short paragraph outlining the potential clinical implications of the observed asymmetry in upper limb use (Page 12, lines 476-484).
“These findings emphasize the need to prioritize rehabilitation strategies that specifically promote the use of the more affected limb. Evidence-based interventions such as constraint-induced movement therapy and bimanual intensive therapy have been shown to enhance motor function and reduce learned nonuse by intensifying practice engaging the more affected arm [47,48]. In addition, incorporating motivational approaches such as playful, goal-directed activities or gamified environments may further encourage spontaneous use of the affected arm in daily life [49,50]. Integrating these approaches could help reduce developmental disregard and promote more balanced use of both arms in children with uCP [49,50].”
17. Although differences in measurement duration between groups are mentioned in the “Limitations” section, there is no discussion of how this might have affected the main conclusions regarding asymmetry and lack of correlation. In children with uCP, the shorter recording period (weekend only) could have reduced activity variability and limited the range of observed behaviours. This issue should be discussed within the interpretation of results, not solely in the limitations section — for instance, whether the correlations might have been weakened by this effect.
Authors’ Response
The manuscript might not have been clear enough about the fact that all between-group comparisons were performed using a 2-day weekend monitoring. The other monitoring periods in TD kids were used only to test whether a 2-day monitoring is representative of a longer monitoring period. We acknowledge that a 2-day monitoring period might be noisier because of factors such as variability across weekends, weather, etc. However, it did not introduce a systematic bias across groups given that the same time window was used for all participants in our statistical analyses. This has now been clarified (Page 8, lines 339-342). Furthermore, previous research has demonstrated that short monitoring periods can provide reliable estimates of motor performance and physical activity in children with CP. For example, Gerber et al. (Ann Phys Rehabil Med, 2021) showed that two consecutive days of sensor-based measurements are reliable for this population's walking performance and physical activity. We have added this reference and clarification to the Methods section (2.3 Study protocol; page 5, lines 215–218) to explicitly justify the use of a shorter recording period. Together with our validation analyses in TD children, this supports the methodological soundness of using a 2-day protocol in the uCP group.
’’ All between-group comparisons were based on the 2-day weekend monitoring data. The longer monitoring periods (5 weekdays and 7 days) available for the TD group were used solely for secondary analyses assessing the reproducibility of weekend measurements relative to longer durations.”
“Moreover, previous research has shown that short monitoring periods of two consecutive days can yield reliable estimates of daily physical activity and gait performance in children with CP, with intraclass correlation coefficients (ICC) ranging from 0.70 to 0.98 when using wearable inertial sensors [30].”
18. The authors rightly observe the lack of association between manual ability (JTHFT), performance (accelerometry), and participation (CAPE), but the interpretation is limited to differences in measurement types. It should be added that such a lack of correlation may also stem from non-linear or indirect relationships (e.g. mediated by psychosocial factors, perceived competence, or family support). A short note on the need for multivariate or mediation analyses (e.g. SEM, mediation analysis) would strengthen the scientific weight of this conclusion.
Authors’ Response
We have added that the lack of direct correlations may reflect indirect or non-linear relationships mediated by psychosocial factors, and that future studies should apply multivariate or path analyses to investigate these effects (Page 13, lines 526-532).
’’ The lack of direct associations among UL capacity, UL performance, and participation may further suggest the existence of more complex, non-linear interrelationships. All factors, including psychosocial and environmental, such as perceived competence and family support, would possibly affect the ability of motor capacity and performance to translate into real-life engagement (59,60]. Future studies should consider analytical approaches such as structural equation modeling or path analysis to better explain the complex determinants of participation in children with uCP [59,60].”
19. The recommendations regarding new technologies (e.g. sensor gloves) are interesting but not directly derived from the study data. It would be better first to relate them explicitly to a specific limitation of accelerometry (e.g. its inability to assess movement quality) and only then propose the technological alternative.
Authors’ Response
The recommendations section has been revised to explicitly link the proposal of new technologies to a key limitation of accelerometry (Page14, lines 591-602).
’’ Although accelerometry offers important insights into the amount and symmetry of UL use, it does not assess qualitative aspects of movement [72,73]. This drawback points out the necessity for additional technologies capable of capturing fine motor skills in real-world contexts. Advancements in sensor fusion and multimodal motor assessment further expand the potential to overcome accelerometry’s inherent limitations [69]. Merging accelerometers with other sensing technologies, such as gyroscopes, markerless systems, and pressure or force sensors, allows assessment of both the kinematic and kinetic features of UL movement [69]. Sensor-integrated gloves show a potential innovation that provides a thorough evaluation of hand use by accurately monitoring finger movements and object manipulation[74]. The combination of these complementary technologies into unified sensor frameworks has a greater potential for the ecological validity and clinical sensitivity of UL assessments in children with uCP."
20. In addition, the “Conclusions” section lacks a clear statement on how the findings confirm or refute the research hypotheses and what they contribute to existing literature. Instead of a summary, it would be preferable to end with a stronger interpretative statement such as: “Our findings suggest that while accelerometry provides objective insights into daily upper limb activity, participation outcomes in children with uCP depend more on contextual and psychosocial than on motor factors.” – please consider this.
Authors’ Response
The conclusion has been revised as requested (Page 15, lines 627-630).

Round 2
Reviewer 2 Report
Comments and Suggestions for Authors
Dear Authors,
The manuscript addresses an important and underexplored topic—the relationship between upper limb capacity, real-world performance, and leisure participation in children with unilateral cerebral palsy (uCP). The topic aligns well with Sensors’ scope, given its focus on accelerometry and quantitative assessment methods. The study is original and clinically relevant; however, substantial issues concerning methodology, structure, and interpretation must be addressed before it can be considered for publication.
Your study addresses a valuable and clinically relevant topic; however, it requires major revision to meet high scientific and editorial standards. Key improvements should focus on:
-
Enhancing methodological transparency and statistical justification.
-
Reducing redundancy and speculative interpretation.
-
Aligning each section with formal writing standards and IMRyC structure.
-
Strengthening coherence among objectives, results, and conclusions.
Title:
-
The title exceeds recommended length and could be simplified for precision (≤12 words).
-
Consider including MeSH terms such as Cerebral Palsy, Accelerometry, and Upper Extremity.
-
Avoid redundant phrasing (“performance and capacity”) unless justified by distinct constructs.
Abstract and Keywords:
-
The abstract is coherent but too descriptive. It should follow IMRyC explicitly:
-
Introduction: briefly outline the rationale and gap.
-
Methods: specify design, participants, measures, and analysis succinctly.
-
Results: emphasize key quantitative findings (means, p-values, effect sizes).
-
Conclusions: restrict to evidence-supported implications.
-
-
Keywords should be standardized to MeSH terms and avoid repetition with title words.
Introduction:
-
The section is lengthy (>900 words). Condense to 400–500 words by removing literature redundancies.
-
Ensure a clear progression from general to specific: burden of CP → concept of capacity/performance/participation → gap in uCP research.
-
The stated hypotheses are appropriate but should be better aligned with the study objectives (primary vs. secondary).
-
Clarify conceptual distinctions using authoritative sources (e.g., WHO ICF).
-
Ensure consistency between hypotheses, results, and conclusions.
Materials and Methods:
-
Design: State explicitly that the study is cross-sectional and observational. Indicate whether data collection occurred prospectively.
-
Participants: Recruitment and inclusion/exclusion criteria are clear; however, indicate how sample size was determined (power analysis).
-
Ethics: Adequate, but specify approval numbers once only.
-
Instrumentation:
-
The transition between devices (ActiGraph → Axivity) is methodologically sensitive. The validation pilot should include exact ICC confidence intervals, bias statistics, and justification for pooling.
-
Provide clearer rationale for selecting a 2-day period in the uCP group.
-
Define data-processing thresholds for non-wear time and activity classification.
-
-
Statistical analysis:
-
Detail handling of missing data.
-
Justify the choice of ANCOVA with small sample size; consider reporting effect sizes with confidence intervals.
-
Include a power analysis (post hoc or a priori).
-
The decision to use Pearson correlation despite potential non-normality should be supported or replaced with non-parametric alternatives.
-
-
Overall, while technically sophisticated, this section requires conciseness and focus on reproducibility.
Results:
-
Presentation is generally structured but somewhat verbose.
-
Start with descriptive data, then main inferential results. Avoid repetition between text, tables, and figures.
-
Include sample size in all analyses and clarify missing data.
-
Tables must follow journal formatting (three horizontal lines). Figures should be self-explanatory with complete legends.
-
Report all statistical outputs (F, p, ηp², CI) consistently.
-
Provide the observed power for non-significant correlations to support interpretation.
Discussion:
-
The discussion is conceptually rich but overly speculative and lengthy.
-
Focus on:
-
Interpretation of main findings relative to the hypotheses.
-
Comparison with existing literature (synthesize, do not list).
-
Clinical and methodological implications.
-
Limitations and future research.
-
-
Avoid extensive explanations on neuroplasticity and behavioral mechanisms unless supported by your data. Replace them with more cautious interpretations.
-
Strengthen internal coherence: ensure that conclusions stem directly from presented results.
-
Reorganize paragraphs logically (from primary to secondary findings).
-
Avoid repetition of results or introduction content.
Conclusions:
-
Should concisely answer the initial objectives, without restating all findings.
-
Remove speculative phrases (“might depend more on contextual factors”) unless empirically justified.
-
Limit to 3–5 sentences summarizing evidence-based implications for research and clinical practice.
Formal and Stylistic Issues
-
Use impersonal, precise scientific English; avoid colloquial or anthropomorphic expressions (e.g., “children may adopt mechanisms”).
-
Ensure coherence of tenses (past for methods/results; present for general statements).
-
Verify that all abbreviations are defined at first mention and consistent throughout.
-
Reference formatting partially follows journal style but should be checked carefully (uniform punctuation, DOI inclusion).
-
Revise manuscript length (ideally ≤7000 words) by tightening redundancy.
With these modifications, the manuscript could make a meaningful contribution to pediatric neurorehabilitation and sensor-based functional assessment research.
Kind regards :)
Author Response
Dear Editor and Reviewers,
We thank you for your thorough and constructive feedback on our manuscript “Upper Limb Capacity, Performance, and Leisure Participation in Children with Unilateral Cerebral Palsy.” We have carefully revised the paper to improve methodological clarity, structure, and coherence between objectives, results, and conclusions.
All the reviewer’s comments were addressed while highlighting the required changes within the manuscript (in yellow). A point-by-point response (outlined in italics and blue) was provided to each reviewer’s comments followed, if applicable, by the extracts from the revised manuscript file with tracked changes (outlined in red and italics) as well as their location in the file (using page and line numbers).
Reviewer(s)' Comments to Author: Reviewer 2
Dear Authors,
The manuscript addresses an important and underexplored topic—the relationship between upper limb capacity, real-world performance, and leisure participation in children with unilateral cerebral palsy (uCP). The topic aligns well with Sensors’ scope, given its focus on accelerometry and quantitative assessment methods. The study is original and clinically relevant; however, substantial issues concerning methodology, structure, and interpretation must be addressed before it can be considered for publication.
Your study addresses a valuable and clinically relevant topic; however, it requires major revision to meet high scientific and editorial standards. Key improvements should focus on:
- Enhancing methodological transparency and statistical justification.
- Reducing redundancy and speculative interpretation.
- Aligning each section with formal writing standards and IMRyC structure.
- Strengthening coherence among objectives, results, and conclusions.
Authors’ Response
We sincerely thank the reviewer for the thoughtful and constructive feedback. The manuscript has been thoroughly revised to enhance methodological transparency, improve structure and coherence, and ensure alignment with scientific and editorial standards as suggested.
Title:
1) The title exceeds recommended length and could be simplified for precision (≤12 words).
Authors’ Response
The title has been shortened and revised to: “Upper Limb Capacity, Performance, and Leisure Participation in Children with Unilateral Cerebral Palsy.”
2) Consider including MeSH terms such as Cerebral Palsy, Accelerometry, and Upper Extremity.
Authors’ Response
Cerebral Palsy is included in the title, and the MeSH terms Accelerometry and Upper Extremity have been added to the keywords.
3) Avoid redundant phrasing (“performance and capacity”) unless justified by distinct constructs.
Authors’ Response
The terms performance and capacity are intentionally both used, as they represent distinct constructs according to the International Classification of Functioning, Disability and Health (ICF): capacity refers to what an individual can do in a standardized environment, while performance reflects what they actually do in daily life. This is explained in the paper’s introduction (Page 2, lines 58-65).
- Abstract and Keywords:
4) The abstract is coherent but too descriptive. It should follow IMRyC explicitly:
-
- Introduction: briefly outline the rationale and gap.
- Methods: specify design, participants, measures, and analysis succinctly.
- Results: emphasize key quantitative findings (means, p-values, effect sizes).
- Conclusions: restrict to evidence-supported implications.
Authors’ Response
The abstract follows the IMRyC structure: the first sentences introduce the rationale and gap, followed by concise descriptions of the methods, key quantitative results with statistical indicators, and evidence-based conclusions. Given the word limit (200-word limit) and the general aspect of the comment, it is difficult to see which specific aspect could be improved.
5) Keywords should be standardized to MeSH terms and avoid repetition with title words.
Authors’ Response
The keywords have been revised to align with MeSH terminology and to avoid redundancy with title terms (Page 2, lines 43-45).
‘’Keywords:
Neurodevelopmental Disorders; Accelerometry; Upper Extremity; Motor Activity; Activities of Daily Living; Child.’’
6) Introduction:
- The section is lengthy (>900 words). Condense to 400–500 words by removing literature redundancies.
- Ensure a clear progression from general to specific: burden of CP → concept of capacity/performance/participation → gap in uCP research.
- The stated hypotheses are appropriate but should be better aligned with the study objectives (primary vs. secondary).
- Clarify conceptual distinctions using authoritative sources (e.g., WHO ICF).
- Ensure consistency between hypotheses, results, and conclusions.
Authors’ Response
We have made some reorganization of the content and somewhat shorten the Introduction. However, we would like to point out that there is no words limit in the journal format. Moreover, given that Sensors is a technoogy-oriented journal and that our paper covers clinical applications, we believe that it is important to cover the different concepts enough for a diverse readership, although some aspects might appear obvious for someone with a clinical background.
Materials and Methods:
7) Design: State explicitly that the study is cross-sectional and observational. Indicate whether data collection occurred prospectively.
Authors’ Response
The study design has been clarified to explicitly state that it is cross-sectional and observational, with data collected prospectively (Page 3, lines 126,130).
‘’This cross-sectional observational study was approved by the institutional review board [RIS board: #2020-1961, #2023-2623, and #2023-2684 Centre intégré universitaire de santé et de services sociaux de la Capitale-Nationale (CIUSSS-CN)] in Québec, and the legal guardian of each child provided written informed consent prior to participation. Data for each group (children with TD and with uCP) were acquired prospectively in distinct studies, which accounts for some differences in the data collection described below.”
8) Participants: Recruitment and inclusion/exclusion criteria are clear; however, indicate how sample size was determined (power analysis).
Authors’ Response
A post-hoc sensitivity power analysis was added in the Results section to report the achieved statistical power based on the final sample size (n = 21 for uCP; n = 30 for TD), and this addition is also mentioned in the Limitations section.
Results section (Page 8, lines 309-312): “A post-hoc sensitivity power analysis conducted in G*Power (version 3.1.9.4 ; α = .05, two-tailed, 1−β= .80) based on the final sample sizes (n = 21 for the uCP group and n = 30 for the TD group) indicated that the study was powered to detect between-group effects of Cohen’s d ≈ 0.81 (ηp2 ≈ 0.14), corresponding to large effect sizes.”
Limitations section (Page 12, lines 490-492): A post-hoc power analysis confirmed that, with the available sample size, the study was sufficiently powered to detect only large effects (Cohen’s d ≈ 0.81), suggesting that smaller or moderate effects might not have been captured.
8) Ethics: Adequate, but specify approval numbers once only.
Authors’ Response
The ethical approval numbers are correctly reported and appear twice only because the journal requires them to be stated both in the Study Design and Ethics section of the manuscript and again in the Institutional Review Board Statement, in accordance with the journal’s guidelines.
- Instrumentation: 9) The transition between devices (ActiGraph → Axivity) is methodologically sensitive. The validation pilot should include exact ICC confidence intervals, bias statistics, and justification for pooling.
Authors’ Response
The validation pilot reports the exact ICC value, 95% confidence interval, bias statistics, and limits of agreement in the revised Methods section (see page 4, lines 173-182). These data confirm the methodological comparability of both devices (ActiGraph GT9X Link and Axivity AX3), supporting their pooling for analyses.
10) Provide clearer rationale for selecting a 2-day period in the uCP group.
Authors’ Response
The rationale for selecting a 2-day monitoring period in the uCP group is detailed in the Study Protocol section (see page 4, lines 148-159). This duration was chosen to minimize participant burden within a longitudinal study involving multiple assessment timepoints. Moreover, previous research (Gerber et al., 2021) has demonstrated that two consecutive days provide reliable estimates of daily physical activity in children with cerebral palsy (ICC = 0.70–0.98).
11) Define data-processing thresholds for non-wear time and activity classification.
Authors’ Response
Non-wear periods and sleep were handled through visual inspection to ensure data quality. This is stated in the Methods (see page 5, lines 202-205). No activity classification thresholds were applied, since the study focused on total activity counts per minute as a global indicator of upper limb performance rather than on activity intensity categories.
Statistical analysis:
12) Detail handling of missing data.
Authors’ Response
Four TD participants were excluded because their average wear time was below 6 hours per day across the two weekend days. This is mentioned in the Sample description (section 3. 1, page 7, lines 288-289). In the remaining participants, there were no missing data for the main variables included in the analyses; therefore, no data imputation was required. This has been added at page 7, lines 294-298, where the average accelerometers wear time by group is also reported.
13) Justify the choice of ANCOVA with small sample size; consider reporting effect sizes with confidence intervals.
Authors’ Response
ANCOVA was selected to control for the covariate age, which is known to influence upper limb activity. All model assumptions (normality, homogeneity, linearity) were verified and met. Effect sizes and their 95% confidence intervals are reported in the Results section to illustrate the magnitude and precision of observed effects (see page 8, lines 312-323).
14) Include a power analysis (post hoc or a priori).
Authors’ Response
A post-hoc sensitivity power analysis was added in the Results section to report the achieved statistical power based on the final sample size (n = 21 for uCP; n = 30 for TD), and this addition is also mentioned in the Limitations section.
Results section (Page 8, lines 309-312): “A post-hoc sensitivity power analysis conducted in G*Power (version 3.1.9.4 ; α = .05, two-tailed, 1−β= .80) based on the final sample sizes (n = 21 for the uCP group and n = 30 for the TD group) indicated that the study was powered to detect between-group effects of Cohen’s d ≈ 0.81 (ηp2 ≈ 0.14), corresponding to large effect sizes.”
Limitations section (Page 12, lines 490-492): A post-hoc power analysis confirmed that, with the available sample size, the study was sufficiently powered to detect only large effects (Cohen’s d ≈ 0.81), suggesting that smaller or moderate effects might not have been captured.
15) The decision to use Pearson correlation despite potential non-normality should be supported or replaced with non-parametric alternatives.
Authors’ Response
This clarification was added in the Statistical Analysis section (see page 6, lines 263-265). Specifically, we noted that JTHFT scores did not meet the normality assumption (Shapiro–Wilk p < 0.05); therefore, all correlations were recomputed using Spearman’s rho. The Results and Table 2 were updated accordingly, without changing the overall interpretation, as correlations between upper limb capacity, performance, and leisure participation remained non-significant.
Page 6, lines 263-265: “Spearman’s rank correlation analyses were performed, as JTHFT scores for both dominant and non-dominant sides did not meet the assumption of normality (Shapiro–Wilk p < 0.05).”
16) Overall, while technically sophisticated, this section requires conciseness and focus on reproducibility.
Authors’ Response
The Methods section has been carefully revised to improve conciseness and readability while maintaining sufficient technical detail to ensure full reproducibility.
Results:
17) Presentation is generally structured but somewhat verbose.
Authors’ Response
The Results section was carefully reviewed and streamlined to improve clarity and conciseness while preserving all essential information.
18) Start with descriptive data, then main inferential results. Avoid repetition between text, tables, and figures.
Authors’ Response
The presentation of results follows this structure, with descriptive data reported first, followed by inferential analyses. Efforts were made to shorten the presentation of results in the text.
19) Include sample size in all analyses and clarify missing data.
Authors’ Response
Sample sizes have now been specified for each analysis in the Results section (see page 10, lines 353-354, lines 370-371). No missing data were present, as all included participants provided complete datasets for the variables analyzed. Four TD participants were excluded a priori due to insufficient wear time (<6 h/day), and all remaining data (n = 21 uCP; n = 30 TD) were complete (see page 7, lines 294-295). For the secondary reproducibility analyses, 28 TD participants with valid 7-day recordings were included.
- Section 3.2.3 (Reproducibility)
Page 10, lines 353-354: “Analyses were conducted on the 28 TD participants who completed valid 7-day recordings.”
- Section 3.2.4 (Effect of age)
Page 10, lines 370-371: “This analysis included all participants (n = 21 uCP; n = 30 TD) with complete valid data.”
20) Tables must follow journal formatting (three horizontal lines). Figures should be self-explanatory with complete legends.
Authors’ Response
All tables were reformatted according to the journal’s guidelines, and figure legends were revised to ensure they are clear and self-explanatory.
21) Report all statistical outputs (F, p, ηp², CI) consistently.
Authors’ Response
All statistical results (F values, p values, partial ηp², and confidence intervals) are now reported consistently throughout the Results section (see page 8, lines 312-323).
22) Provide the observed power for non-significant correlations to support interpretation.
Authors’ Response
A post-hoc power analysis was added in the Results section to report the achieved statistical power (≈0.06–0.28) for the observed non-significant correlations (see page 9, lines 338-341). This information was also acknowledged in the Limitations section, emphasizing that non-significant results should be interpreted with caution due to the limited sample size (see page 12, lines 492-396).
Results section (see page 9, lines 338-341): A post-hoc power analysis indicated that, with the sample size of 21 participants in the uCP group, the study had limited statistical power (approximately 0.06–0.28) to detect small-to-moderate correlations between upper limb performance, capacity, and participation intensity.
Limitations section (see page 12, lines 492-396): The study’s small sample size resulted in limited statistical power for detecting moderate correlations between upper limb performance, capacity, and participation intensity, as confirmed by the post-hoc analysis. Non-significant results should therefore be interpreted with caution.
23) Discussion:
- The discussion is conceptually rich but overly speculative and lengthy.
- Focus on:
- Interpretation of main findings relative to the hypotheses.
- Comparison with existing literature (synthesize, do not list).
- Clinical and methodological implications.
- Limitations and future research.
- Avoid extensive explanations on neuroplasticity and behavioral mechanisms unless supported by your data. Replace them with more cautious interpretations.
- Strengthen internal coherence: ensure that conclusions stem directly from presented results.
- Reorganize paragraphs logically (from primary to secondary findings).
- Avoid repetition of results or introduction content.
Authors’ Response
We have revised the discussion to shorten it and avoid repetitions. However, note that the Discussion section was substantially revised after the first round of review to address prior reviewer suggestions. Specifically, the explanations regarding neuroplasticity and behavioral mechanisms were added following reviewers’ recommendations to better contextualize the findings. Nevertheless, these sections were carefully written to remain concise and directly linked to the study results. The current version maintains a balanced interpretation while respecting the reviewers’ earlier input.
24) Conclusions:
- Should concisely answer the initial objectives, without restating all findings.
- Remove speculative phrases (“might depend more on contextual factors”) unless empirically justified.
- Limit to 3–5 sentences summarizing evidence-based implications for research and clinical practice.
Authors’ Response
The Conclusion section was revised to address these points while respecting the reviewers’ earlier input.
Formal and Stylistic Issues
25) Use impersonal, precise scientific English; avoid colloquial or anthropomorphic expressions (e.g., “children may adopt mechanisms”).
Authors’ Response
The manuscript was revised for precise and impersonal scientific language (see page 1, lines 19-20 and page 10, lines 392-394).
26) Ensure coherence of tenses (past for methods/results; present for general statements).
Authors’ Response
Tenses were revised for consistency between sections.
27) Verify that all abbreviations are defined at first mention and consistent throughout.
Authors’ Response
All abbreviations have been verified to ensure that they are defined at first mention and used consistently throughout the manuscript.
28) Reference formatting partially follows journal style but should be checked carefully (uniform punctuation, DOI inclusion).
Authors’ Response
References were carefully revised to ensure full compliance with the journal’s formatting style, including punctuation and DOI inclusion.
29) Revise manuscript length (ideally ≤7000 words) by tightening redundancy.
Authors’ Response
The manuscript was revised and reduced from 5757 to 5536 words. However, the journal guidelines do not specify a word limit.
Reviewer 3 Report
Comments and Suggestions for Authors
After the re-review, I have no further comments, congratulations on the work.
Author Response
Reviewer(s)' Comments to Author: Reviewer 3
After the re-review, I have no further comments, congratulations on the work.
Authors’ Response
We sincerely thank the reviewer for the positive feedback and appreciation of our work.